

# Performance of the Adriatic Sea and Coast (AdriSC) climate component – a COAWST V3.3-based coupled atmosphere-ocean modelling suite: atmospheric part

Cléa Denamiel[1,2], Petra Pranić[1], Damir Ivanković[1], Iva Tojčić[1], Ivica Vilibić[1]

[1]Institute of Oceanography and Fisheries, Šetalište I. Meštrovića 63, 21000 Split, Croatia
[2]Ruđer Bošković Institute, Division for Marine and Environmental Research, Bijenička cesta 54, 10000 Zagreb, Croatia

*Correspondence to*: Cléa Denamiel (cdenami@irb.hr)
ORCiD: 0000-0002-5099-1143

**Abstract.** In this evaluation study, the coupled atmosphere-ocean Adriatic Sea and Coast (AdriSC) climate model, which was implemented to carry out 31-year long evaluation and climate projection simulations in the Adriatic and northern Ionian seas, is briefly presented. The kilometre-scale AdriSC atmospheric results, derived with the Weather Research and Forecasting (WRF) 3-km model for the 1987-2017 period, are then thoroughly compared to a comprehensive publicly and freely available observational dataset. The evaluation shows that overall, except for the summer surface temperatures which are systematically underestimated, the AdriSC WRF 3-km model has a far better capacity to reproduce the surface climate variables (and particularly the rain) than the WRF regional climate models at 0.11° of resolution. In addition, several spurious data have been found in both gridded products and *in situ* measurements which thus should be used with care in the Adriatic region for climate studies at local and regional scales. Long-term simulations with the AdriSC climate model, which couples the WRF 3-km model with a 1-km ocean model, might thus be a new avenue to substantially improve the reproduction, at the climate scale, of the Adriatic Sea dynamics driving the Eastern Mediterranean thermohaline circulation. As such it may also provide new standards for climate studies of orographically-developed coastal regions in general.

## 1 Introduction

In the past decade, within the climate community, scientific efforts led by the COordinated Regional climate Downscaling EXperiment (CORDEX; Giorgi et al., 2009) facilitated the rapid development and applications of Regional Climate Models (RCMs) around the world (e.g. Rinke etal. 2011; Nikulin et al., 2012; da Rocha et al., 2014; Huang et al., 2015; Ruti et al., 2016; Zou and Zhou, 2017; Di Virgilio et al., 2019). However, RCMs often fail to represent extreme events as, for example, they do not properly resolve complex orography, coastline, and land-sea contrasts (Prein et al., 2015). Consequently, the need to study climatic hazards and their extremes with kilometre-scale atmospheric models has recently been promoted (e.g. summer 2020 call for CORDEX Flagship Pilot Study, https://cordex.org/wp-content/uploads/2020/07/FPS-flyer-summer2020.pdf). Additionally, in coastal regions, such atmospheric models should be coupled with high-resolution ocean models in order to





quantify the impact of these extreme conditions on the ocean dynamics and, therefore, on the marine ecosystems, the erosion or the transport of pollutants, or other. But, due to their prohibitive numerical cost, coupled atmosphere-ocean kilometre-scale climate models are not commonly used in long-term climate research.

[Figure 1]

Nevertheless, over the elongated semi-enclosed Adriatic basin (Figure 1), only high-resolution limited-area models can represent the atmosphere-ocean interactions during extreme events (e.g. Pasarić et al., 2007; Prtenjak et al., 2010; Ricchi et al., 2016; Cavaleri et al., 2010, 2018). The complex geomorphology of the Adriatic Sea indeed includes (a) more than 1200 islands, islets, ridges and rocks mostly located in the northeast, (b) mountains surrounding the entire basin and (c) bathymetries
evolving from a really shallow and wide shelf in the north to a deep pit in the south. Additionally, orographically-driven extreme windstorms in the northern Adriatic (i.e. the so-called bora winds; Brzović and Strelec Mahović, 1999; Grisogono and Belušić 2009) are known to strongly influence the annual dense water budget in the Adriatic Sea as well as interannual to decadal thermohaline and biogeochemical variability between the Adriatic and the northern Ionian seas (Roether and Schlitzer, 1991; Gačić et al., 2010; Bensi et al., 2013; Batistić et al., 2014). The Adriatic Sea and Coast (AdriSC) kilometre-scale climate
model (Denamiel et al., 2019) was thus recently developed to represent the long-term atmospheric and oceanic circulations in the Adriatic basin, with the perspective of future applications to extreme event hazard assessment, ecosystem modelling, sediment and larvae transport, and others. Furthermore, for climate projections, the pseudo-global warming (PGW) downscaling method (Schär et al., 1996) was proven to greatly improve the future projections of precipitations and convective storms in atmospheric kilometre-scale climate models (Prein et al., 2015). Consequently, this method was first extended to
coupled atmosphere-ocean models, then implemented in the AdriSC climate component, and finally tested successfully with an ensemble of short-term climate simulations for wind-driven extreme events in the Adriatic Sea (Denamiel et al., 2020a, 2020b). The need to use kilometre-scale models during severe bora events for proper representation of the Adriatic long-term thermohaline circulation was also demonstrated (Denamiel et al., 2021).

Following these preliminary results and the high-resolution studies implemented in other parts of the world (e.g. Chan et al.,
2018; Li et al., 2019; Knist et al., 2020), a 31-year long evaluation run was performed with the AdriSC climate model for the 1987-2017 period. The presented work assesses the skill of the AdriSC atmospheric kilometre-scale model while the evaluation of the AdriSC ocean coastal model is done separately. Contrarily to global or regional climate models, the evaluation of kilometre-scale models requires the use of observational datasets with high temporal resolution (i.e. at least hourly in the atmosphere and daily in the ocean) and spatial coverage (i.e. network of *in situ* measurements or kilometre-scale gridded
products) for both atmospheric and oceanic essential climate variables. But, such datasets are intrinsically uncertain and therefore not entirely reliable. For example, (a) ground-based station measurements often present inhomogeneities due to change in instruments or environmental conditions, (b) long-term time series are difficult to obtained from measurements at sea which highly depend on the ship and buoy locations and (c) remote sensing data generally have low temporal resolution





(i.e. daily), do not measure directly the climate variables and can include systemic disturbances due to the impact of the atmosphere. Moreover, based on the assumption that the quality of the observational datasets can be assessed with climate models, Massonnet et al. (2016) highlighted the need to provide guidance for a more objective selection of the observations used in evaluation studies. Another key point concerning the atmospheric observational datasets, and most particularly the *in situ* measurements, is the ease (and cost) of access which highly depends on the data sharing policy of the different providers.

For example, collecting ground-based station data from the different meteorological agencies around the Adriatic basin requires to contact each provider separately (e.g. Italy, Croatia, Montenegro, Albania, etc.) and in many cases, implies receiving, after a long delay, partial datasets provided in different formats. Consequently, only open access datasets accessible on the web and provided in an easily readable format, are used hereafter for the evaluation of the AdriSC climate component. The inconvenient of this choice is that, due to, for example, unit conversions and rounding errors, the quality of the datasets can be sometimes degraded before being shared online.

The following study will thus be, as suggested by Massonnet et al. (2016), a bidirectional exercise evaluating both the kilometre-scale AdriSC atmospheric model and the freely available observations retrieved, in the Adriatic basin, from *in situ* measurements, gridded datasets and remote-sensing products. The AdriSC modelling suite (i.e., in more details, its climate component, its web portal and the set-up of its atmospheric model) as well as the observations and methods used to perform the skill assessment of the model are first presented in Section 2. Then, Section 3 displays and discusses the results consisting

in basic, spatially distributed statistical/seasonal and vertical skill assessments, as well as, in the comparisons of measured and modelled climatologies and distributions. Finally, the findings of this study are summarized in Section 4.

## 2 Model, data and methods

### 2.1 Modelling suite

#### 2.1.1 AdriSC climate component

The Adriatic Sea and Coast (AdriSC) modelling suite (Denamiel et al. 2019) has been developed with the aim to accurately represent the processes driving the atmospheric and oceanic circulation at different temporal and spatial scales over the Adriatic and northern Ionian seas (Figure 1). For climate studies, the AdriSC climate component (Table 1) is set-up to provide kilometre-scale hourly results for 31-year long simulations. The evaluation run covering the 1987-2017 period is partially presented in this study. The far-future runs (2070-2100 period) are derived with the Pseudo-Global Warming (PGW)

methodology recently extended to coupled atmosphere-ocean models (Denamiel et al., 2020a) and tested for an ensemble of short-term extreme events in the Adriatic Sea (Denamiel et al., 2020a, 2020b). In this climate configuration (Table 1), the Adriatic atmospheric processes, depending on both local orography and Mediterranean regional forcing, are represented with a 3-km grid (266 x 361) encompassing the entire Adriatic and northern Ionian seas (Figure 1, top right panel). Additionally, the AdriSC 3-km grid is nested in a 15-km outer grid (horizontal size: 140 x 140) approximately covering the central





Mediterranean basin. In the ocean, the exchanges of the Adriatic Sea with the Ionian Sea are captured with a 3-km grid identical to the atmospheric domain, while an additional nested 1-km grid (676 x 730) represents more accurately the complex geomorphology of the Adriatic Sea. The vertical discretization of the grids is achieved via terrain-following coordinates: 58 levels refined in the surface layer for the atmosphere (Laprise, 1992) and 35 levels refined near both the sea surface and bottom floor for the ocean (Shchepetkin, 2009). Additionally, a Digital Terrain Model (DTM) incorporating offshore bathymetry from

ETOPO1 (Amante and Eakins, 2009), nearshore bathymetry from navigation charts CM93 2011, topography from GEBCO 30 arc-second grid 2014 (Weatherall et al., 2015) and coastline data generated by the Institute of Oceanography and Fisheries (Split, Croatia), is providing the high-resolution orography, bathymetry and coastline of all the AdriSC grids.

The AdriSC climate model is based on a modified version of the Coupled Ocean-Atmosphere-Wave-Sediment-Transport (COAWST V3.3) modelling system developed by Warner et al. (2010). The state-of-the-art COAWST model couples (online)

the Regional Ocean Modeling System (ROMS svn 885) (Shchepetkin & McWilliams, 2009) and the Weather Research and Forecasting (WRF v3.9.1.1) model (Skamarock et al., 2005) via the Model Coupling Toolkit (MCT v2.6.0) (Larson et al., 2005) and the remapping weights were computed – between the WRF 15-km, WFR and ROMS 3-km and ROMS 1-km atmospheric and ocean grids, with the Spherical Coordinate Remapping and Interpolation Package (SCRIP). Within the AdriSC climate model, the COAWST model is compiled with the Intel 17.0.3.053 compiler, the PNetCDF 1.8.0 library and

the MPI library (mpich 7.5.3) on the European Centre for Middle-range Forecast's (ECMWF's) High Performance Computing Facility (HPCF). In addition, ecFlow 4.9.0 – the work flow package used by all ECMWF operational suites, is set-up to automatically and efficiently run the AdriSC long-term simulations in a controlled environment. In terms of workload, no hyper-threading is used and the AdriSC climate model optimally runs on 260 CPUs, with both the WRF and ROMS grids decomposed in 10 x 13 tiles (Denamiel et al., 2019). Despite this optimal configuration of the models which maximizes the

running time of each individual model as well as the time used to exchange data between the different grids, the AdriSC climate model runs at extreme computational cost and about 18 months are needed to complete each 31-year long simulation within the ECMWF HPCF.

[Table 1]

### 2.1.2 AdriSC climate component web portal

Storage and accessibility of climate model results is known to be challenging even at the regional scale. With the kilometre-scale coupled atmosphere-ocean AdriSC climate component, more than 245 TB of raw data is generated for each 31-year long simulation and safely stored on the ECMWF tape system (i.e. ECFS). However, this storage is not easily accessible and post-processed hourly 2D and daily 3D atmospheric (i.e. WRF 3-km) and oceanic (i.e. ROMS 3-km and ROMS 1-km) fields (representing about 7 TB of data per 31-year long simulation) are available on a local Network-Attached Storage (NAS) server

(ftp://messi-nas.izor.hr/AdriSC). Given the numerical cost associated with running the AdriSC climate component, user-





friendly and efficient extraction and analysis of the model results is crucial for the dissemination to the scientific community in broad. The AdriSC climate web portal (https://vrtlac.izor.hr/ords/adrisc/interface_form) is thus designed to easily retrieve the model results in time and space – i.e. horizontally at a given pressure or depth, vertically along a transect and at a given point, and generate NetCDF files and/or figures, depending on the demands of the users (Figure 2).

130                                                   [Figure 2]

### 2.1.3 Atmospheric model set-up

The full description of the AdriSC climate component requires a detailed presentation of both the atmospheric and oceanic kilometre-scale model set-up. However, as the AdriSC evaluation is performed in two parts, only the set-up of the AdriSC WRF 3-km model – solely used to force the AdriSC ROMS 3-km and ROMS 1-km grids, is briefly presented in this study.

The atmospheric model physics and parametrizations, set-up in the AdriSC WRF 3-km model, are based on the optimal configuration of Adriatic high-resolution WRF models described by Kehler-Poljak et al. (2017): Morison 2 moment scheme microphysics scheme (Morrison et al., 2005), MYJ Planetary Boundary Layer (Janjić, 1994), Dudhia (Dudhia, 1989) and RRTM (Mlawer et al., 1997) short and longwave radiation schemes, Eta surface layer scheme (Janjić, 1994) and Five-layer thermal diffusion scheme for soil temperature (Dudhia, 1996). Additionally, for the evaluation run (Table 1), the initial
conditions and boundary forcing of the WRF 15-km grid are provided by the 6-hourly ERA-Interim reanalysis fields (Balsamo et al. 2015). Finally, as the spatial extension of the ocean grids does not entirely cover the WRF 15-km atmospheric domain, the Sea Surface Temperature (SST) from the ROMS grids is not prescribed to the WRF models. This approach avoids any potential discontinuities along the border between the two-way nested WRF 15-km and WRF 3-km atmospheric grids and optimizes the balance between the AdriSC model efficiency and accuracy by reducing the exchanges between the different
grids. The SST forcing is thus provided by the Mediterranean Forecasting System (MFS) high resolution (1/16° x 1/16°) MEDSEA re-analysis (Simoncelli et al., 2014), also used as boundary conditions for the ROMS 3-km grid.

### 2.2 Skill assessment

### 2.2.1 Observations

In this study, the AdriSC WRF 3-km model performance is assessed for 6 different variables (i.e. temperature, dew point, rain,
pressure and wind speed and direction) by comparison to a comprehensive collection of freely available observational data retrieved for the 1987-2017 period from *in situ* measurements, gridded datasets and remote-sensing products.

The first product included in this observational collection, is the E-OBS (v21.0e) ensemble dataset (https://surfobs.climate.copernicus.eu/dataaccess/access_eobs.php). It is continuously provided via the Copernicus and Climate change service initiatives and updated every year as more data become available (Cornes et al., 2018). The data



consists in 0.1° regular grids of mean daily surface temperature, accumulated daily precipitations (referred as daily rain hereafter) and daily mean sea-level pressure over the land. E-OBS is a European climate monitoring product based on surface *in situ* observations collected by ground-based observation networks (mostly owned and operated by the National Meteorological Services) and derived from a 100-member ensemble of conditional simulations. E-OBS is thus widely used to evaluate atmospheric regional climate models over the land, particularly by the EURO-CORDEX community.

As nearly half of the AdriSC WRF 3-km domain is at sea, two remote-sensing products are also used in this evaluation. On the one hand, the Cross-Calibrated Multi-Platform or CCMP V2 (Atlas et al., 2011; Mears et al., 2019), continuously provides 6-hourly gridded surface wind speed and direction over the sea at 0.25° resolution for the 1987-2017 period (http://www.remss.com/measurements/ccmp/). It is derived via a Variational Analysis Method from the combination of (a) Version-7 RSS radiometer wind speeds, QuikSCAT and ASCAT scatterometer wind vectors, (b) moored buoy wind data, and

(c) ERA-Interim model wind fields. On the other hand, the gridded daily accumulated precipitations (referred as daily rain hereafter) over the sea at 0.25° resolution are derived from the 3-hourly Tropical Rainfall Measuring Mission (TRMM) Multi-Satellite Precipitation Analysis TMPA (3B42). TRMM is provided by the NASA GES DISC (Huffman et al., 2007) for the 1998-2017 period (https://disc.gsfc.nasa.gov/datasets/TRMM_3B42_Daily_7/summary).

Finally, instead of low spatial and temporal resolution gridded products, *in situ* observations have been directly used in this

evaluation. Within the AdriSC WRF 3-km domain, the available measurements of about 350 ground-based stations recorded during the 1987-2017 period are easily accessible from the Integrated Surface Database (ISD) hosted by the National Oceanographic and Atmospheric Agency (NOAA). This dataset is hereafter referred as NOAA stations (https://gis.ncdc.noaa.gov/maps/ncei/cdo/hourly). It includes hourly observations of 10 m wind speed and direction, 2 m temperature, 2 m dew point, sea-level pressure as well as accumulated 6-hourly surface precipitation (referred as 6-hourly rain

hereafter) compiled from different meteorological agencies and provided with a common user-friendly ASCII format. Even though a systemic and automatic quality-check (QC) is already applied before the integration of the observations in the ISD, a second more thorough manual QC was done for each variable (except the rain) of the extracted NOAA stations, in order to remove duplicated stations, obvious outliers and bad data. The QC of the rain would have required to track each individual storm during the 1987-2017 period and was thus not undertaken. At the end, 251 NOAA stations were kept for the evaluation

of the AdriSC WRF 3-km model (Figure 1). Additionally, in order to evaluate the vertical structure of the atmospheric model, soundings taking twice per day (at 00:00 and 12:00 UTC) and available at four different locations – i.e. Rome and Udine in Italy as well as Zadar and Zagreb in Croatia (Figure 1) – are also extracted during the 1987-2017 period from the database of the University of Wyoming (UWYO; http://weather.uwyo.edu/upperair/sounding.html).

A full list of the data collected to perform the AdriSC WRF 3-km model evaluation during the 1987-2017 period is presented

in Table 2. The table includes, for each of the five datasets (i.e. NOAA stations, E-OBS, CCMP, TRMM and UWYO





soundings), the observed variables, the height and the frequency at which the measurements are taken and the total number of records.

[Table 2]

### 2.2.2 Methods

Once the evaluation run is completed, the extraction of the AdriSC WRF 3-km model hourly results is achieved either via bilinear interpolation to the coarser coordinates of the E-OBS, CCMP and TRMM gridded products with the Earth System Modelling Framework (ESMF) software or via a near-neighbour method at points in time and space matching the coordinates of the *in situ* observational datasets (i.e. NOAA stations and UWYO soundings). For the UWYO soundings the AdriSC WRF 3-km results are also linearly interpolated to the vertical structure of the measurements following the height.

The evaluation of the AdriSC WRF 3-km model skill is following several steps. First, the results are evaluated in the form of a Taylor diagram (Taylor, 2001), a robust method to visualize multiple statistical parameters within a single plot and perform a basic assessment of the model behaviour. Second, the bias or difference between model results and observations for each variable separately, is calculated at each point in time and space of the E-OBS, CCMP, TRMM and NOAA station datasets. The biases are analysed in space with statistical quantities such as median (or mean for the rain) and Median (Mean for rain)

Absolute Deviation (MAD) as well as $1^{st}$, $25^{th}$, $75^{th}$ and $99^{th}$ percentiles. In this study, in order to obtain more robust statistics for the chosen geophysical quantities which are likely to be heavy tailed due to extreme conditions, the use of median and MAD is preferred to the mean and standard deviation preconized for normal distributions. However, despite having a heavy tailed distribution, the rain is not a continuous quantity – i.e. occurrences of rain in the Adriatic region are low and the median is likely to be close to zero. Consequently, the mean and Mean Absolute Deviation are used for the statistical analysis.

Additionally, the bias for the NOAA stations is also analysed seasonally for each variable separately – with winter defined as December January February (DJF), spring as March April May (MAM), summer as June July August (JJA) and autumn as September October November (SON). This analysis is used to better identify the spatial and seasonal behaviour of the model depending on the different variables. Then, the daily climatology (daily median and MAD or, for the extreme rain, $98^{th}$, $99.5^{th}$ and $99.9^{th}$ percentiles), the density probability function and the apparent scaling rate (i.e. linear relationship between the logarithm of the extreme precipitations and the 2 m temperatures; Drobinski et al., 2018) of both model results and observations

are compared for the entire NOAA station dataset in order to assess the capacity of the model to reproduce the overall observed daily climatology, hourly distributions and extreme precipitations. Finally, the capacity of the AdriSC WRF 3-km model to reproduce the observed vertical structure is presented as the median of the bias between model and soundings for the temperature, dew point, pressure and wind speed at Rome, Udine, Zadar and Zagreb locations between the surface and 15 km of height (interpolated every 10 m till 5 km and then every 1000 m) for the entire 1987-2017 period.






## 3 Results and Discussions

### 3.1 Basic skill assessment

In the atmosphere, accurate representation of the orography is crucial for mesoscale climate modelling. Thus, for the observations located in land (i.e. E-OBS and NOAA stations), the first assessment of the AdriSC WRF 3-km model simply

consists in looking at the differences in elevation between measurements and model (Figure 1, bottom panels). For both the NOAA stations and the E-OBS gridded product these differences are mainly lower than 20 m, except along the Apennines, the Dinaric Alps and the Hellenides where they can reach up to 300 m. For the NOAA stations, the AdriSC WRF 3-km model orography seems to overall strongly underestimate the elevation along the Apennines (up to 300 m) while overestimated it along the Dinaric Alps (by in average 100 m). However, it should be noticed that the NOAA station locations are extracted

from the 3-km model results with a near-neighbour methodology – i.e. the closest point of the grid is picked without interpolation. Consequently, these differences do not necessarily imply that the orography used in the AdriSC WRF 3-km model is inaccurate. It simply shows that the location of the extracted point may have a spatial offset of 1 to 2 km compared to the station position. For the E-OBS product, the alternation of strongly positive and negative elevation differences (±150 m in average) along all the mountains (i.e. Apennines, Dinaric Alps and Hellenides) shows that most probably the orography used to produce the E-OBS dataset is far smoother than the one used in the AdriSC WRF 3-km model. These differences in

used to produce the E-OBS dataset is far smoother than the one used in the AdriSC WRF 3-km model. These differences in orography may have some important consequences concerning certain physical processes like precipitations along the mountains.

[Figure 3]

Another basic assessment of the AdriSC WRF 3-km model is presented with a Taylor diagram. It illustrates - for each variable

(i.e. temperature, dew point, rain, pressure, wind speed and wind direction) and each dataset (i.e. E-OBS, CCMP, TRMM, NOAA stations and UWYO soundings), the correlation and the normalized standardized deviations between model and observations (Figure 3). For all the datasets, the worst statistics (correlations lower than 0.5 and normalized standardized deviations of around 0.5 or above 1.5) are obtained for the rain which appears to be poorly captured by the AdriSC WRF 3-km model. Independently of the variable, the best statistics (correlation above 0.9 except for the wind direction and normalized

standardized deviations near 1) are reached for the UWYO soundings. This potentially shows a good representation of the atmospheric vertical structure by the AdriSC WRF 3-km model. Finally, better statistics (higher correlations and normalized standardized deviations closer to unity) are always obtained with the hourly NOAA station measurements than with the gridded daily or 6-hourly products (i.e. E-OBS, CCMP and TRMM). This is, to some extent, surprising as climate models generally better reproduce the observations at daily than hourly scale and for smoother spatial results than at precise station positions.

Even though Taylor diagrams are extremely useful for basic skill assessment of regional climate models depending on various datasets, they may not thus be precise enough to properly evaluate the hourly results of kilometre-scale models.





### 3.2 Spatially distributed statistical skill assessment

For E-OBS daily temperature, rain and pressure in land (Figures 4 to 6 and supplementary material Figure S1), for CCMP 6-hourly wind speed and direction at sea (Figures 7, 8 and supplementary material Figure S2) and for TRMM daily rain at sea
(Figure 9 and supplementary material Figure S2), spatial maps of the median (or mean for the rain) and MAD of the gridded observations as well as the median (or mean for the rain), MAD and 1st, 25th, 75th, 99th percentiles of the biases between the AdriSC WRF 3-km results and the observations, are analysed.

[Figure 4]

For the surface temperature at 2 m height in coastal areas (Figure 4 and supplementary material Figure S1), the median of the
E-OBS data shows important spatial variations with temperature reaching (a) 15.0 °C (±6.5 °C of variations derived from the MAD) in average (up to 18.0±5.5 °C) in the south along the Italian coast, (b) 13.0-14.0±6.0 °C in average along the Croatian coast and (c) up to 18.0±6.0 °C along the Montenegrin, Albanian and Greek coasts. In land, temperatures are lower with in average 10.0±5.5 °C (down to 7.0±5.5 °C) along the Apennines and 7.0±6.5 °C (down to 5.0±6.5 °C) along the Dinaric Alps and the Hellenides. Additionally, in the Pannonian plain, temperatures reach 12.0-13.0±7.0 °C. Concerning the evaluation, the
AdriSC WRF 3-km model is overall largely underestimating the temperatures with a negative median bias of -1.5 °C in the mountains (±1.5 °C in the Apennines, ±2.5 °C in the Dinaric Alps and ±2.0 °C in the Hellenides), -3.5±1.6 °C along the Adriatic coast and down to -5.0±3.2 °C in the Pannonian plain and the Po valley. In terms of extreme conditions, the negative bias reaches down to -4.0 °C in the mountains, -6.0 °C along the coast and -8.0 °C in the Pannonian plain for the 25th percentile and down to -7.0 °C in the mountains, -10.0 °C along the coast and -15.0 °C in the Pannonian plain for the 1st percentile.
Concerning the extreme overestimations of the AdriSC WRF 3-km temperatures, on the one hand, the positive bias only reaches up to 1.0-2.0 °C for mountains particularly the Dinaric Alps for the 75th percentile – with most of the domain still having a negative bias of about 1.0-2.0 °C. On the other hand, it reaches up to 8.0 °C in the mountains, 5.0 °C along the coast and 6.0 °C in the Pannonian plain for the 99th percentile. The AdriSC WRF 3-km model is thus incapable to accurately reproduce the highest surface temperatures captured in land with the E-OBS dataset along the Adriatic. These results are
following the work of Varga and Breuer (2020) who, for a 1-year long period, studied the sensitivity of simulated 2 m temperature to different WRF 10-km configurations over a domain which partially cover the Adriatic basin. Specifically, they found that, for any WRF configuration, the spatial distributions of the annual mean temperature bias relative to the E-OBS dataset present a general underestimation of about -4.0 to -3.0 °C.

[Figure 5]

For the daily accumulated rain in land (Figure 5 and supplementary material Figure S1), the mean of the E-OBS data reveals that the strongest precipitations occur mostly along the Croatian coast in the lee of the Dinaric Alps and in Istria as well as along the Apennines – in average 3.0±3.5 mm/day and up to 5.0±5.0-8.0 mm/day. The AdriSC WRF 3-km model tends to





overestimate the daily rain at these very places (as well as in southern Italy and in Greece) with a positive mean bias reaching 1.0±3.0 mm/day in average and up to 2.0±9.0 mm/day. In the rest of the domain the mean bias is slightly negative and reaches

0.5±2.5 mm/day in average. Concerning the extreme biases, they reach (a) -1.5 mm/day extremely locally in northern Croatia for the 25th percentile, (b) -10.0 mm/day on average (up to -40.0 mm/day) over the entire domain for the 1st percentile, (c) less than 1.5 mm/day along the mountains for the 75th percentile and (d) up to 100.0 mm/day along the coastal mountains for the 99th percentile. These results present in fact a great improvement compared to the WRF models used within the EURO-CORDEX RCM ensemble (i.e. European domain of the CORDEX community). Indeed, Kotlarski et al. (2014) found that the

EURO-CORDEX's WRF models were overestimated the mean E-OBS precipitations by more than 100 % over most of the Adriatic region (for both summer and winter) while the biases of the AdriSC WRF 3-km only vary between -40 % in the northern Italy and 50 % along the lee of the highest mountains. Additionally, given the 0.1° resolution of the E-OBS dataset, a smoother orography than for the AdriSC WRF 3-km model is used to extrapolate the observed rainfall (Figure 1). Consequently, the precipitation differences highlighted by the statistical spatial skill assessment (i.e. up to 50 % along the

highest mountains) do not necessarily imply that the model is inaccurate and more analyses are needed to reach a definite conclusion.

[Figure 6]

For the daily sea-level pressure in land (Figure 6 and supplementary material Figure S1), an obvious defect of the E-OBS dataset is exhibited by the spatial variations of the median reaching 1024.0-1026.0±6.0-6.5 hPa along the Montenegrin coast

and radiating towards the Dinaric Alps and southern Italy with values decreasing to 1018.0 hPa. Similarly, along the northern Croatian coast, another area reaches 1021.0±8.0 hPa at around 45°North of latitude. These problems thus cast a doubt on the accuracy of the E-OBS sea-level pressure over the entire eastern part of the AdriSC WRF 3-km domain as well as in southern Italy. Consequently, the pressure bias is only analysed for the northern Italian peninsula. For this area, the AdriSC WRF 3-km model tends to overestimate the sea-level pressure with a positive median bias below 2.0±2.6-3.2 hPa (1.0±2.8 hPa in average)

while the extreme biases reach -2.0 hPa for the 25th percentile, -10.0 hPa for the 1st percentile, 4.0 hPa for the 75th percentile and 10.0 hPa for the 99th percentile. Within the EURO-CORDEX ensemble, Kotlarski et al. (2014) found that, over the entire European domain, the WRF winter wet biases seemed closely related to distinct negative biases of mean sea-level pressure, indicating too-high intensity of low pressure systems passing the continent. From the limited results analysed here, it thus seems that the AdriSC WRF 3-km is better suited to represent the low pressure systems over the Adriatic region than these

RCMs.

[Figure 7]

For the 6-hourly wind at sea (Figures 7, 8 and supplementary material Figure S2), the median of the CCMP wind speed shows that the strongest winds occur in the northern Ionian, southern Adriatic and Tyrrhenian seas reaching 5.5±2.0-2.5 m/s in





average, while the median wind direction is mostly 240-280±40-80 °North (i.e. degree North following meteorological
conventions). However, in the middle and northern Adriatic, the median of the wind speed is lower than 4.5±2.0 m/s, with a
median wind direction of 80-160±90-125 °North. The median wind speed bias is positive over the Adriatic Sea and reaches
2.5±1.7 m/s in the northern Adriatic and along the Croatian coast. It is associated with wind direction biases of the order of 0-
20±20-40 °North. However, it is extremely small (absolute bias below 0.2±1.2-1.7 m/s in average, except near the coasts) over
the northern Ionian and Tyrrhenian seas, where the median of the wind direction biases is mostly -30-0±20-40 °North. In terms
of extremes, on the one hand, the under estimation of the wind speed by the AdriSC WRF 3-km model reaches -1.7 m/s over
the northern Ionian and Tyrrhenian seas for the 25[th] percentile and down to -8.0 m/s over the northern Ionian and the Adriatic
seas for the 1[st] percentile. On the other hand, the overestimation reaches 4.5 m/s in the northern Adriatic Sea for the 75[th]
percentile and up to 15.0 m/s also in the northern Adriatic Sea for the 99[th] percentile. Concerning the wind direction bias, the
extreme values can reach up to ±360 °North for the 1[st] and 99[th] percentiles, which probably reveals that time correlation of the
observed and model wind directions can be low. For the 75[th] percentile, the AdriSC WRF 3-km model overestimates the CCMP
wind direction by 60-80 °North along the Croatian coastline and the northern Adriatic Sea, where the wind speed is also largely
overestimated for both the 75[th] and 99[th] percentiles. However, the CCMP products are known to underestimate high wind
speed events (> 25.0 m/s). Additionally, the bora – a northern to north-eastern downslope wind associated with speeds of 20.0-
30.0 m/s (Grisogono and Belušić, 2009) – is regularly blowing in these littoral areas mostly during winter and spring.
Therefore, the overestimation of the wind speeds and directions, particularly in the northern Adriatic, may be linked to the
CCMP product and not the inaccuracy of the AdriSC WRF 3-km model.

[Figure 8]

Finally, for the daily accumulated rain at sea (Figure 9 and supplementary material Figure S2), the median of the TRMM data
highlights that the heaviest rain (up to 5.0±5.0-9.0 mm/day) is falling along the northern Croatian coast and the south-eastern
Adriatic coast. The mean of the rain bias reveals that the AdriSC WRF 3-km model tends (a) to overestimate slightly the rain
(0.5-1.0±5.0-9.0 mm/day) along the Italian coastline of the Adriatic Sea and (b) to underestimate it by up to 1.5-2.0±3.0-4.0
mm/day along the eastern coast of the Adriatic Sea as well as at the boundaries of the WRF 3-km model. This boundary effect
is linked to the fact that the WRF 3-km model which resolves some of the small-scale convective clouds, is nested into the
coarser WRF 15-km domain for which the Kain-Fritch cumulus parameterization (Kain, 2004) is used. Concerning the extreme
precipitations, on the one hand, the negative bias is quasi-null over the entire domain for the 25[th] percentile and reaches up to
-70 mm/day along the eastern Adriatic coast for the 1[st] percentile. On the other hand, the positive bias is up to 1 mm/day along
the Italian coast for the 75[th] percentile and up to 40 mm/day over the entire Adriatic Sea for the 99[th] percentile. However,
Kolios and Kalimeris (2020) have shown that the TRMM monthly product (3B43) is characterized by
an overestimation tendency over the northern and higher altitude regions of the central Mediterranean, including the Adriatic
Sea. They also found that heavy rainfall episodes are underestimated over the marine Mediterranean regions by this product.



[Figure 9]

In brief, the statistical spatial skill assessment of the AdriSC WRF 3-km model against E-OBS, CCMP and TRMM products has revealed some important discrepancies between the climate model results and the observations. In terms of median (mean for the rain) and MAD over the entire domain, on the one hand, the model underestimation reaches up to 5.0±3.2 °C for the daily surface land temperatures and 2.0±4.0 mm/day for the daily rain at sea. On the other hand, the model overestimation reaches up to (a) 2.0±9.0 mm/day for the daily land rain, (b) 2.0±3.2 hPa for the daily land sea-level pressure and (c) 2.5±1.7 m/s for the 6-hourly wind speed at sea. However, except for the temperature, some questions can be raised concerning the quality of these daily (or 6-hourly) gridded products over the Adriatic and northern Ionian domain. Additionally, previous studies (e.g. Bauer et al., 2011; Prein et al., 2013; Warrach-Sagi et al., 2013) have shown that the added value of atmospheric kilometre-scale models compared to RCMs, can cancel out by spatial and temporal averaging. Consequently, a more precise assessment of the AdriSC WRF 3-km model skills should be done by direct comparison with the ground-based NOAA stations which provide more reliable observations and can be more easily quality-checked (except for the rain).

**3.3 Spatially distributed seasonal skill assessment**

In this sub-section, the biases between the AdriSC WRF 3-km hourly (6-hourly for the rain) results and the ground-based atmospheric observations are seasonally analysed, at each of the 251 NOAA stations, with the median and MAD values for the 2 m temperature, 2 m dew point, sea-level pressure, surface rain, 10 m wind speed and 10 m wind direction (Figures 10 to 15 and supplementary material Figures S3 to S8).

[Figure 10]

The evaluation of the hourly temperature biases (Figure 10 and supplementary material Figure S3) confirms and refines the conclusions reached in Section 3.2. First, the AdriSC WRF 3-km model is capable to capture the observations during winter (DJF) with a good accuracy: median values varying between -0.75 °C and +0.75 °C over the entire domain. However, a dozen stations show extreme values reaching -2.50 °C and +2.50 °C, and MAD values below ±1.75 °C along the coast and above ±2.50 °C in the mountains, similarly to the other seasons (not analysed further). Then, the model shows no skill to represent the extreme temperatures during summer (JJA), when the median bias is below -3.50 °C for the entire domain, except for a few stations where it surprisingly tends to zero. Finally, for spring (MAM) and autumn (SON), the median temperature bias is mostly negative over the entire domain with values varying between -3.00 °C and -0.50 °C, except at some stations along the coast and in the Apennines and Dinaric Alps where it can be slightly positive with values mostly below 1.00 °C. These results are somehow aligned with the previous study of Kotlarski et al. (2014) evaluating climate regional models using WRF at 12-km resolution. In this study, all the WRF models showed a cold bias over the Adriatic basin during summer (about -3.00 to -2.00 °C), but also during winter (also about -3.00 to -2.00 °C) due to a problem of snowfall and snow cover (Mooney et





al. 2013; García-Díez et al. 2015). It is thus interesting to point out that the use of the AdriSC WRF 3-km model largely improves the results in winter but seems to slightly increase the negative biases in summer.

[Figure 11]

Curiously, the AdriSC WRF 3-km shows better skills to overall capture the observed dew point at the NOAA stations (Figure
11 and supplementary material Figure S4). In winter, the model slightly overestimates the dew point with median biases below 1.00±1.50 °C, except along the coast and in the mountains where they can reach up to 3.00±2.50 °C and down to -3.00±2.00 °C, respectively. In spring, the model seems to even better represent the dew point with positive median biases below 0.75±2.00 °C over the entire domain, except at a few stations where either the median biases reach up to 2.50±2.00 °C or down to -2.50±1.75 °C. In summer and autumn, the model keeps overestimating the dew point along the coast with median bias of about
1.00±2.00 °C (up to 4.00±2.50 °C) in summer and 0.75±1.75 °C (up to 4.00±2.25 °C) in autumn. Additionally, the model underestimates the dew point in the mountains and the plains with median bias of about -1.50±2.00 °C (down to -4.00±2.50 °C) in summer and -1.25±1.75 °C (down to -4.00±2.00 °C) in autumn. As the extreme temperatures are underestimated by the AdriSC WRF 3-km model, particularly in summer, the relatively small bias obtained for the dew point implies that the model is also lacking of accuracy concerning the relative humidity, which is likely to be overestimated following the approximation
from Lawrence (2005).

[Figure 12]

Contrarily to the previous results obtained with E-OBS daily rain and the Taylor diagram, the AdriSC WRF 3-km model seems capable to capture the observed 6-hourly rain at the NOAA stations (Figure 12 and supplementary material Figure S5), with a good accuracy independently of the season. Indeed, the absolute mean bias is always below 0.25±0.75 mm/day over the entire
domain, except at some locations mostly along the Italian coast where it can reach 2.50±5.00 mm/day. However, even though these results are encouraging, the mean and MAD values are not representative of the model capacity to reproduce extreme rain events for which higher percentiles should be used for further analysis.

[Figure 13]

Concerning the sea-level pressure evaluation which could not be thoroughly performed with the E-OBS dataset, the AdriSC
WRF 3-km model shows a good agreement with the NOAA station quality-checked observations (Figure 13 and supplementary material Figure S6). The best results are obtained in winter when the absolute median bias is below 0.75±0.80 hPa (up to 5.00±1.80 hPa), with a slight overestimation of the model, except for some stations in the mountains. The strongest overestimation of the pressure over the entire domain is however found in summer, with a median bias of 3.00-5.00±0.80-1.80 hPa, except in the Apennines and some stations in the coastal Dinaric Alps where it tends to zero. For both spring and autumn,
the AdriSC WRF 3-km model tends to overestimate the sea-level pressure for the entire domain, with a median bias below





1.00±0.80-1.80 hPa (up to 4.00±1.80 hPa), except for a few stations where the bias is slightly negative (above -0.75±1.80 hPa). It is worth noticing that, independently of the seasons, the MAD tends to be nearly null along the mountain peaks, small along the coast and higher in the Pannonian plain.

[Figure 14]

Concerning the wind (Figures 14 and 15 and supplementary material Figures S7 and S8), the overall performance of the AdriSC WRF 3-km model seems satisfactory, independently of the seasons. Indeed, the median speed and direction biases tend towards 0.0 m/s and 0 °North for nearly half of the NOAA stations. Extreme values are found along the coast, over the Po valley and in the Pannonian plain and reach up to 1.5-2.0±1.5-2.5 m/s for the wind speed and ±30 °North (with a viability of 25-45 °North) for the wind direction. Additionally, despite a strict quality-check of the wind measurements which has

eliminated all the stations known to be in sheltered positions – such as, for example, three coastal stations (Senj, Rab and Mali Lošinj) in Croatia (Belušić and Klaić, 2004; Klaić et al., 2009; Belušić et al., 2013; Kuzmić et al., 2015) – the systematic overestimation of the wind speed by the climate model at certain stations may still be linked to some problems with the observations.

[Figure 15]

To summarize, the seasonal analysis of the biases between the AdriSC WRF 3-km model and the NOAA stations has highlighted some important results concerning the skills of the AdriSC WRF 3-km model over the land. First, the fact that the model shows no skill to capture the highest temperatures in summer is confirmed. Second, contrarily to the previous results, the 6-hourly rain seems to be accurately enough represented by the model concerning the mean and MAD values, yet the extremes (e.g. 98th to 99.9th percentiles) should also be checked. Third, the atmospheric pressure is relatively well described

by the model over the entire domain, even though slightly overestimated, particularly in summer. And finally, the wind speed and direction are found to be reproduced by the model, despite a systematic overestimation at certain stations.

### 3.4 Skill assessment via climatology and distribution comparisons

In this section, the differences between the AdriSC WRF 3-km hourly (6-hourly for the rain) results and the ground-based atmospheric observations are first analysed as daily climatology over the full AdriSC WRF 3-km domain and for the entire set

of the 251 NOAA stations. The median and associated variabilities (i.e. median ± MAD representing the upper and lower bounds) are used for the 2 m temperature, 2 m dew point, sea-level pressure, 10 m wind speed and 10 m wind direction, while the 98th, 99.5th and 99.9th percentiles are chosen in order to represent the extreme surface rain (Figures 16 and 17).

[Figure 16]





For the temperature, dew point and pressure (Figure 16), the daily climatology analysis confirms the results obtained in the

previous sections. First, the AdriSC WRF 3-km model is only capable to represent the 2 m temperature during winter and largely underestimates it by down to -3.0 °C in spring and autumn and by more than -5.0 °C in summer. Similar results were found by Varga and Breuer (2020) for a WRF model using the same physics than the AdriSC WRF 3-km but coarser horizontal (10-km) and vertical (31 levels) resolutions, particularly in summer (i.e. biases down to -5.4 °C in July) but also for all the other seasons (i.e. bias of -4.5 °C annually). They also demonstrate that the temperature bias can be largely reduce by using

other numerical scheme for the planetary boundary and surface layers than the ones used in their study. Second, the model seems to represent quite accurately the daily climatology of the 2 m dew point, with maximum differences of less than 1.5 °C occurring mostly during the winter, when the model overestimates the measurements. Third, the model is also capable to overall capture the daily climatology of the sea-level pressure, except in summer when it overestimates it by more than 2.0 hPa. Finally, for the 2 m temperature, the 2 m dew point and the sea-level pressure, the daily variability (i.e. upper and lower

bounds) of the AdriSC WRF 3-km model is similar to the one obtained with the entire dataset of the 251 NOAA stations.

For the extreme 6-hourly rainfall (Figure 16), the 98th, 99.5th and 99.9th percentiles of the AdriSC WRF 3-km model overall follow the results obtained with the entire dataset of the 251 NOAA stations. However, for the 163rd, 193rd and 211th days of the year, three peaks recorded by the NOAA stations are not seen by the model. For these three days, between 39 and 56 occurrences of 6-hourly rainfall above 200 mm were observed at the ground-based stations. If extreme rainfall can occur during

severe storms in autumn with rates up to 300 mm/24h (Davolio et al., 2016), it seems improbable that these occurrences cumulate during three specific days in June and July. The three peaks are thus considered to be the result of bad data and are ignored. Concerning the more detailed analysis of the results, the AdriSC WRF 3-km model tends to (a) accurately represent the 98th percentile of the observed 6-hourly rainfall at the daily scale, (b) slightly underestimate (by less than 2 mm/6h) the 99.5th percentile for the entire year, except for some few days in late summer and autumn, and (c) underestimate (by 5 to 20

mm/6h) the 99.9th percentile for the entire year, except for some few days in late summer and autumn when it can overestimate it by up to 40 mm/6h. Overall, this implies that the AdriSC WRF 3-km model is reproducing the extreme rain with a good enough accuracy but not necessarily with the right timing concerning the most extreme events.

[Figure 17]

Finally, concerning daily climatology of the wind speed and direction (Figure 17), the AdriSC WRF 3-km model seems to

overall overestimate the wind speed by up to 2 m/s particularly in winter, while reproducing with a good accuracy the wind direction (slight underestimation by about -15 °North in autumn and winter). However, the daily observed wind speeds, derived from the entire dataset of the 251 NOAA stations, exhibit a non-continuous behaviour with sharp daily changes between one value to the other for both median and upper/lower bounds. On the contrary, the model results are more smoothly transitioning from one day to the other, with obvious seasonal behaviours (e.g. stronger winds in winter) not observed with the NOAA

stations. The quality of the observed wind speeds, including representativeness of station locations, can thus be questioned.





In order to provide an overview of the model behaviour, the probability density functions of the hourly 2 m temperature, 2 m dew point and 10 m wind speed for the entire dataset of the 251 NOAA stations, are also analysed in this sub-section (Figure 17). It is important to notice that the probability density functions are obtained via a kernel smoothing method, which presents the advantage of generating continuous distributions but may overestimate the tails of these distributions. Surprisingly, the

most important results of this analysis are not related to how the AdriSC WRF 3-km model compares to the observations, but to the unexpected shape of the distributions of the observed quantities extracted from the NOAA stations. These distributions indeed exhibit a non-continuous behaviour with sharp changes between one value to the other resulting in multiple peaks (i.e. distributions shaped like hedgehogs). It is important to know that the NOAA station temperature, dew point and speed were provided as integer values in the U.S customary units (i.e. Fahrenheit for temperatures and miles per hour for speed). However,

the data used in this study were all originally collected by European meteorological stations in the metric system units (i.e. degree Celsius and meter per second) and with their own unknown rounding errors. The presented data thus went through two unit conversions. First, they were converted from the metric system to the U.S. customary units with rounding to the closest integer, before integration to the ISD. Second, they were re-converted to degree Celsius and meter per second before being treated in this study. Consequently, the "hedgehog" shape of the observed distributions is most probably resulting from these

accumulated rounding errors and unit conversions. In light of these results, it is important to understand that the biases between the AdriSC WRF 3-km model and the NOAA station observations may generally be overestimated. For example, the 35 % probability of having 3 m/s wind speed in the NOAA station distribution is probably highly exaggerated. Indeed, due to the rounding errors and the unit conversions, this peak is surrounded by underestimation of the other wind speed values (i.e. below 10 % of probability to have 2.25 m/s and 3.75 m/s wind speeds). The 18 % probability obtained with the model results for 3

m/s wind speeds may thus be more realistic as coming from a smooth and continuous distribution. However, the NOAA station dataset still provide valid comparisons concerning the general behaviour of the model. For example, it shows how the highest temperatures are strongly underestimated by up to 10 °C, while the dew points are somewhat better represented except for the extreme values.

[Figure 18]

The last analysis provided in this sub-section is the scaling between precipitation extremes and temperatures. Indeed, under the hypothesis of constant relative humidity, extreme precipitations increase at a scaling rate of 7.00 %/°C following the Clausius-Clapeyron (CC) relationship which plays a key role in climate studies (e.g. Betts and Harshvardhan, 1987; Held and Soden, 2006; Westra et al., 2014). In this study, the observed and modelled apparent scaling rates – derived from the linear relationship between the logarithm of the extreme precipitations (i.e. 99[th] percentiles) and the 2 m temperatures (Drobinski et

al., 2018) – are thus compared seasonally for the dataset of the 251 NOAA stations (Figure 18). First, it should be noticed that the observed apparent scaling rates are always below the CC scaling rate and can even be negative. This is, however, in good agreement with the results found around the Adriatic basin by Drobinski et al. (2018). Second, the apparent scaling rates





extracted from the model results are overall following the tendencies of the observations, independently of the season. Third,
in more details, the scaling rates are reproduced by the AdriSC WRF 3-km (a) the most accurately during winter and summer,
with underestimations below ±0.40 %/°C compared to the observed values of 2.66%/°C and -1.83%/°C, respectively, and (b)
the least accurately during spring and autumn, with an underestimation of 1.61 %/°C (compared to the 1.68 %/°C observed)
and an overestimation of 0.89 %/°C (compared to the 3.18 %/°C observed), respectively. Finally, independently of the seasons,
the amount of extreme precipitations (i.e. 99th percentile) depending on the 2 m temperature tends to always be underestimated
by the AdriSC WRF 3-km model and shifted by at least 5 °C for the lowest temperatures.

In a nutshell, the analysis of the climatologies and distributions has revealed that, except for the summer temperatures at 2 m
height and atmospheric sea-level pressure, the AdriSC WRF 3-km model is overall capable to reproduce the observed
conditions for the ensemble of the 251 ground-based stations selected in this study. Additionally, the distributions of the NOAA
station dataset have been shown to present some non-continuous behaviour most probably linked to the accumulated rounding
errors and unit conversions. Finally, the seasonal apparent scaling rates obtained for the entire Adriatic basin with the AdriSC
WRF 3-km model are generally in good agreement with those obtained with the observations, except during spring.

### 3.5 Vertical skill assessment

The final analysis performed in this study concerns the capability of the AdriSC WRF 3-km model to reproduce the observed
vertical structure. Unfortunately, the ERA5 re-analysis cannot be used for this evaluation, following Denamiel et al. (2021)
who demonstrated that this product is not capable to reproduce bora events in the northern Adriatic. The only products available
for the vertical skill assessment of the AdriSC WRF 3-km model in the Adriatic basin are therefore the four long-term sounding
records extracted from the UWYO database. In this section, the seasonal biases between the AdriSC WRF 3-km model and
the sounding data recorded twice a day at Rome, Udine, Zadar and Zagreb, are thus presented for the temperature, dew point,
pressure and wind speed (Figure 19).

[Figure 19]

First and quite surprisingly, for all the seasons, the vertical behaviour of the AdriSC WRF 3-km model seems to be independent
of the location of the soundings. Indeed, for each variable, the median of the vertical biases is overall similarly distributed for
the four different stations. Second, the temperature biases are the strongest in surface (down to -5.0 °C in summer as seen
before) but tend to rapidly decrease with the height to reach nearly 0.0 °C between 2.5 km and 10 km of height, independently
of the season. Between 10 and 15 km, however, the biases present a fast increase (up to 2.5 °C) till 12 km, followed by a fast
decrease towards 0.0 °C. Third, the dew point biases tend to be small, except in summer, and negative between the surface and
5 km of height – on average between -2.0 and -0.5 °C, except for Udine where they reach -4.0 °C in summer at 2.5 km height.
However, above 5 km of height, they steadily grow up to 6.0-8.0 °C at 15 km of height. Then, independently of the season,
the pressure biases are strong at the surface with values between 0.5 and 2.5 hPa, but quasi null between 1 and 5 km of height





and steadily increasing till up to 3.5 hPa at 12.5 km of height. Finally, the wind speed biases show more variability, depending
525 on the location of the soundings. However, they are overall strong between the surface and 1 km of height with values up to
2.0 m/s, minimum between 1 and 8 km (between -0.5 and 0.5 m/s, except for Zagreb where they reach -1.0 m/s at 2 km) and
quite important between 8 and 13 km with negative values reaching down to -1.5 m/s at different heights, depending on the
location and the season. It should also be mentioned that 7.4 % of the sounding data were recorded below 1 km of height while
33.7 % were taken in the troposphere above 1 km (i.e. 3.7 % per kilometre) and 16.5 % in the stratosphere up to 15 km (i.e.
530 3.3 % per kilometre). The remaining 42.4 % of the sounding data was recorded above 15 km (up to 48 km), beyond the AdriSC
WRF 3-km vertical computational limit. This means that biases up to 1 km of height have been derived with more data than
the ones between 1 and 15 km of height. Following these results, the AdriSC WRF 3-km model seems to present the strongest
biases in the boundary layer and the stratosphere (above 10 km), but to be capable to properly capture the dynamics of the
troposphere above 1 km of height.

## 535  4 Summary and perspectives

In this study, the performance over the Adriatic region of the WRF 3-km model – forcing, within the AdriSC modelling suite,
the ROMS 3-km and 1-km ocean models – has been described in detail for a 31-year long evaluation climate run (i.e. 1987-
2017 period). However, the evaluation of kilometre-scale coupled atmosphere-ocean models – which requires high quality
observations with dense spatial coverage and hourly records – is not yet state-of-the-art in the climate community.
Consequently, the quality of the comprehensive dataset of open source remote sensing and *in situ* observations used in this
study was also discussed at length. Overall, the presented work highlighted two important points. First, the AdriSC WRF 3-
km model demonstrates some skill to represent the climate variables, with the exception of the summer temperatures
systematically underestimated by up to 5 °C over the entire domain. Second, several problems exist over the Adriatic region
concerning the open source observations collected for the evaluation. For example, the E-OBS dataset presents spurious results
of mean sea-level pressure along the eastern Adriatic coast and the quality of the ground-based station records provided by the
NOAA seems to have been degraded due to successive unit conversions and rounding errors leading to non-continuous
distributions (i.e. probability density functions with a hedgehog shape). Despite these limitations, the added value of the
AdriSC WRF 3-km over the Adriatic region has clearly been demonstrated. The use of the AdriSC WRF 3-km model indeed
leads to a better representation of the temperatures (except in summer), the atmospheric pressure and above all the
precipitations compared to the results of the WRF models from the EURO-CORDEX RCM ensemble. Unfortunately, due to
the extremely high computational costs associated with running such coupled atmosphere-ocean kilometre-scale models, the
Mediterranean climate community is still not convinced to further develop them in areas where RCMs are known to fail to
reproduce extreme conditions.



The evaluation of the AdriSC climate model is, in fact, only the first step towards the quantification of the added value of such
kind of models in the Adriatic Sea. For example, no consensus in a unified theory explaining the Adriatic-Ionian Bimodal
Oscillating System (BiOS) – driving substantial inter-annual to decadal thermohaline oscillations in the Adriatic Sea – has yet
been reached within the scientific community. Indeed, the drivers of this process are hypothesized to be either the Adriatic
dense water or the local effects of pressure and/or wind-driven patterns (e.g. Molcard et al., 2002; Borzelli et al., 2009; Gačić
et al., 2010; Pinardi et al., 2015; Reale et al., 2017; Rubino et al., 2020). Consequently, the AdriSC climate model has also
been developed with the aim to expand the knowledge on which driver is most important for modelling the BiOS. In this
context, it was designed to properly capture the orographically-driven severe bora events (Denamiel et al., 2021) occurring in
the northern Adriatic with strong temporal (i.e. hourly) and spatial (i.e. kilometre to sub-kilometre scales) variabilities (Belušić
and Klaić, 2004; Grisogono and Belušić, 2009; Kuzmić et al., 2015). These events are indeed associated with strong sea surface
cooling known to precondition the dense water formation and the thermohaline circulation of the Adriatic Sea (e.g. Artegiani
et al., 1997; Orlić et al., 2007; Janeković et al., 2014; Vilibić et al., 2018; Denamiel et al., 2020b). It is thus expected that the
detailed analysis of the 31-year long AdriSC climate evaluation run will provide, in a near future, more robust and more reliable
results concerning the drivers of the BiOS, but also better representation of the orographically-driven wind storms and their
impact on the ocean processes such as the Adriatic thermohaline circulation. Additionally, the future climate of the bora winds
and the BiOS has been, so far, documented through an assessment of EURO-CORDEX and Med-CORDEX climate models
at 0.11° horizontal resolution (Somot et al., 2006; Belušić Vozila et al., 2019). Consequently, the analysis of the 31-year long
AdriSC projections under climate warning scenario (2070-2100 period) may also provide some new insights concerning the
future of severe bora dynamics as well as of the associated dense water formation and Adriatic thermohaline circulation.

In conclusion, within the new CORDEX framework promoting the use of kilometre-scale models to study the impact of climate
change on extreme events and their long-term consequences, the Adriatic region seems to be a perfect laboratory where to
experiment and develop this new type of approaches.

**Code availability**

The code of the COAWST model as well as the ecFlow pre-processing scripts and the input data needed to re-run the AdriSC
climate model in evaluation mode for the 1987-2017 period can be obtained under the Open Science Framework (OSF) FAIR
data repository https://osf.io/zb3cm/ (doi:10.17605/OSF.IO/ZB3CM).

**Data availability**

The model results and the measurements as well as the post-processing scripts used to produce this article can be obtained
under the Open Science Framework (OSF) FAIR data repository https://osf.io/uex9p/ (doi:10.17605/OSF.IO/UEX9P).



**Author contribution**

IV and CD defined concept and design of the study. Material preparation was done by IT and CD. Set-up of the model and
simulations were performed by CD. Web portal for the AdriSC climate model results was created by DI. Production of the
figures was done by PP and CD. Analysis of the results was performed by IV and CD. The first draft of the manuscript was
written by CD. All authors were engaged in commenting, revising and polishing of the manuscript. All authors read and
approved the final manuscript.

**Competing interests**

The authors declare that they have no conflict of interest.

**Acknowledgments**

The contribution of all the organisations that provided the observations used in this study – the Copernicus and Climate change
service initiatives at https://marine.copernicus.eu, the National Oceanic and Atmospheric Administration (NOAA) at
https://www.ncdc.noaa.gov, the National Aeronautics and Space Administration (NASA) at
https://disc.gsfc.nasa.gov/datasets/TRMM_3B42_Daily_7/summary, the Remote Sensing System (RSS) at
http://www.remss.com/measurements/ccmp/ and the University of Wyoming (UWYO) at
http://weather.uwyo.edu/upperair/sounding.html – is acknowledged. Acknowledgement is also made for the support of the
European Centre for Middle-range Weather Forecast (ECMWF) staff, in particular Xavier Abellan and Carsten Maass, as well
as for ECMWF's computing and archive facilities used in this research. This work has been supported by projects ADIOS
(Croatian Science Foundation Grant IP-2016-06-1955), BivACME (Croatian Science Foundation Grant IP-2019-04-8542),
CHANGE WE CARE (Interreg Croatia-Italy program) and ECMWF Special Project (The Adriatic decadal and inter-annual
oscillations: modelling component).

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



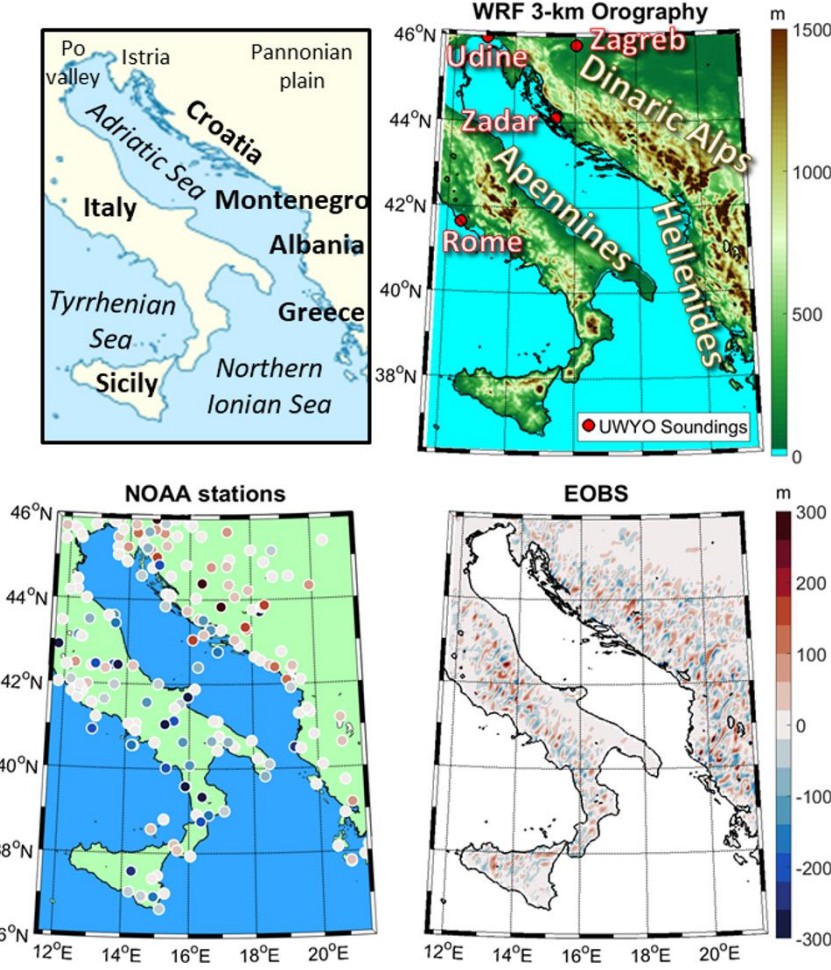

**Figure 1. Name of the geographical (top left panel) and topographical (top right panel) features of the AdriSC WRF 3-km model domain, location of the UWYO soundings (top right panel) and biases between the AdriSC WRF 3-km orography and both the NOAA stations (bottom left panel) and the E-OBS dataset (bottom right panel) elevations.**





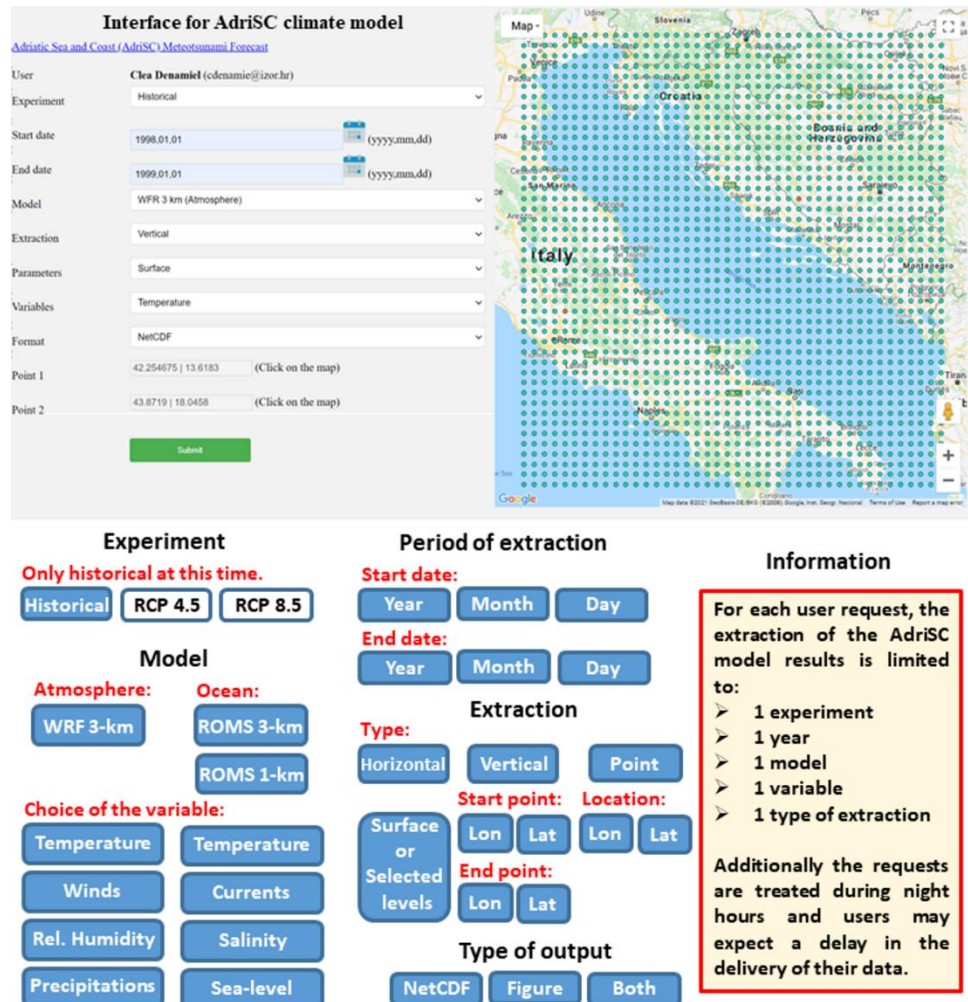

**Figure 2.** Interface for the extraction of the AdriSC climate model results available at https://vrtlac.izor.hr/ords/adrisc/interface_form (top panel) and schematic representation of the available options of extraction (bottom panel).





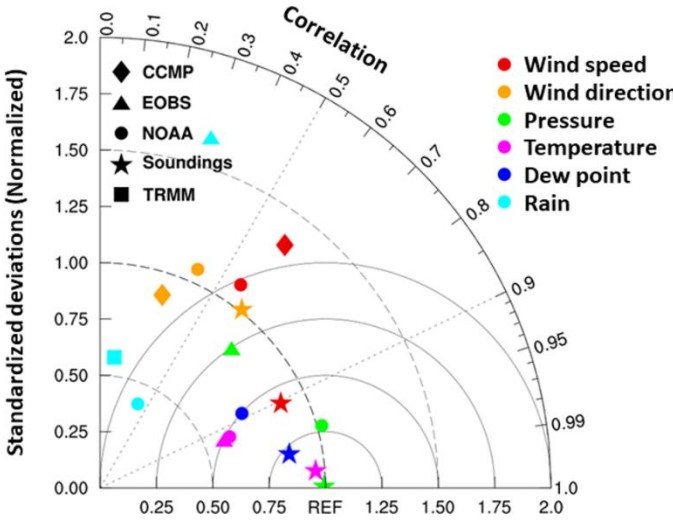

**Figure 3. Taylor diagram summarizing the overall skills of the AdriSC WRF 3-km model to reproduce wind speed and direction, sea-level pressure, temperature, dew point and rain compared to freely available observations including the E-OBS gridded dataset, the CCMP and TRMM remote-sensing gridded products and the NOAA ground-based stations and UWYO soundings *in situ* measurements.**





**Figure 4. Median of the E-OBS daily mean temperature dataset over the land (top left panel) as well as median (top right panel) and 25th (centre left panel), 75th (centre right panel), 1st percentile (bottom left panel), 99th (bottom right panel) percentiles of the daily temperature biases between AdriSC WRF 3-km model results and E-OBS dataset over the land during the 1987-2017 period.**





**Figure 5.** Mean of the E-OBS daily rain dataset over the land (top left panel) as well as mean (top right panel) and 25[th] (centre left panel), 75[th] (centre right panel), 1[st] (bottom left), 99[th] (bottom right panel) percentiles of the daily rain biases between AdriSC WRF 3-km model results and E-OBS dataset over the land during the 1987-2017 period.



**Figure 6. As in Fig. 3 but for the E-OBS daily mean pressure over the land.**







**Figure 7.** As in Fig. 3 but for the 6-hourly CCMP wind speed remote sensing data over the sea.



**Figure 8. As in Fig. 3 but for the 6-hourly CCMP wind direction remote sensing data over the sea.**



**Figure 9. As in Fig. 3 but for the daily TRMM rain remote sensing data over the sea during the 1998-2017 period.**



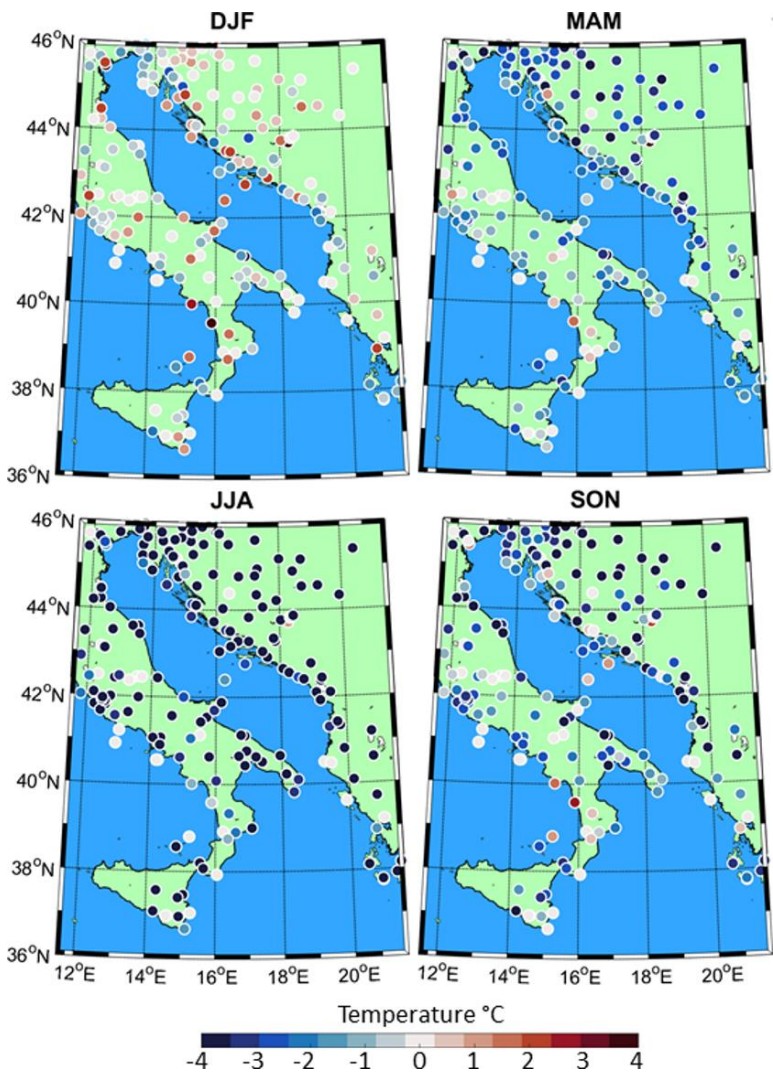

835

**Figure 10. Seasonal variations of the median hourly temperature bias between AdriSC WRF 3-km model results and NOAA land station measurements during winter (DJF), spring (MAM), summer (JJA) and autumn (SON) for the 1987-2017 period.**

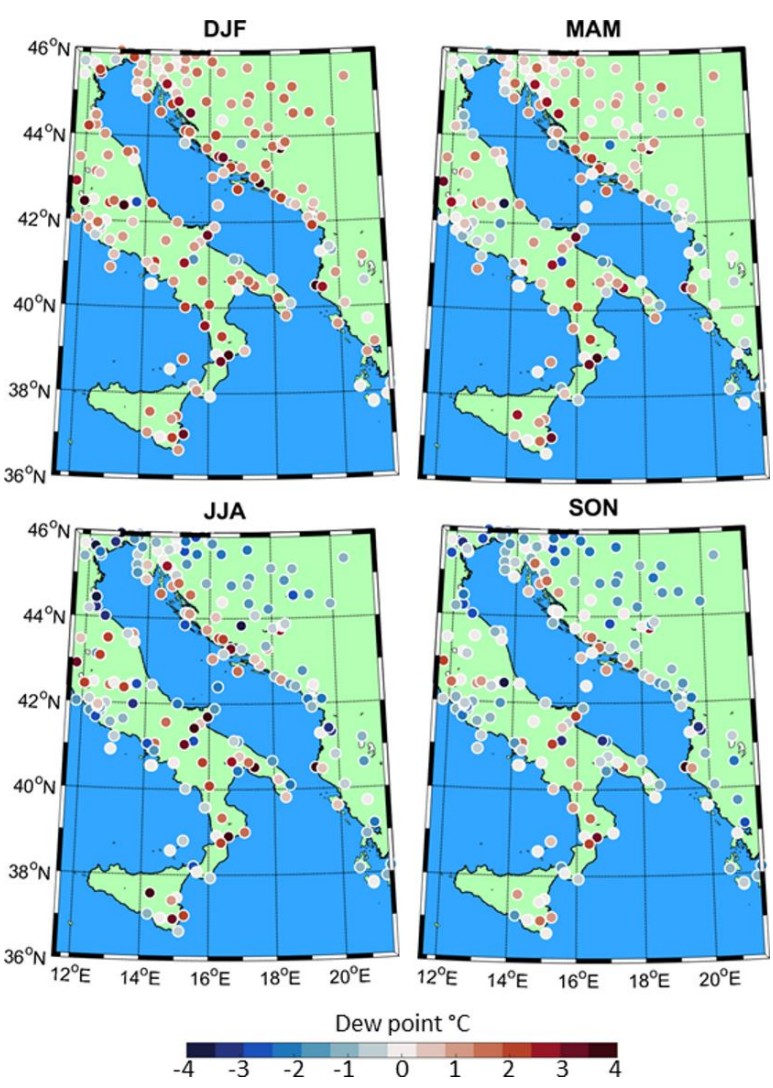

**Figure 11. As in Fig. 9 but for the hourly dew point bias.**





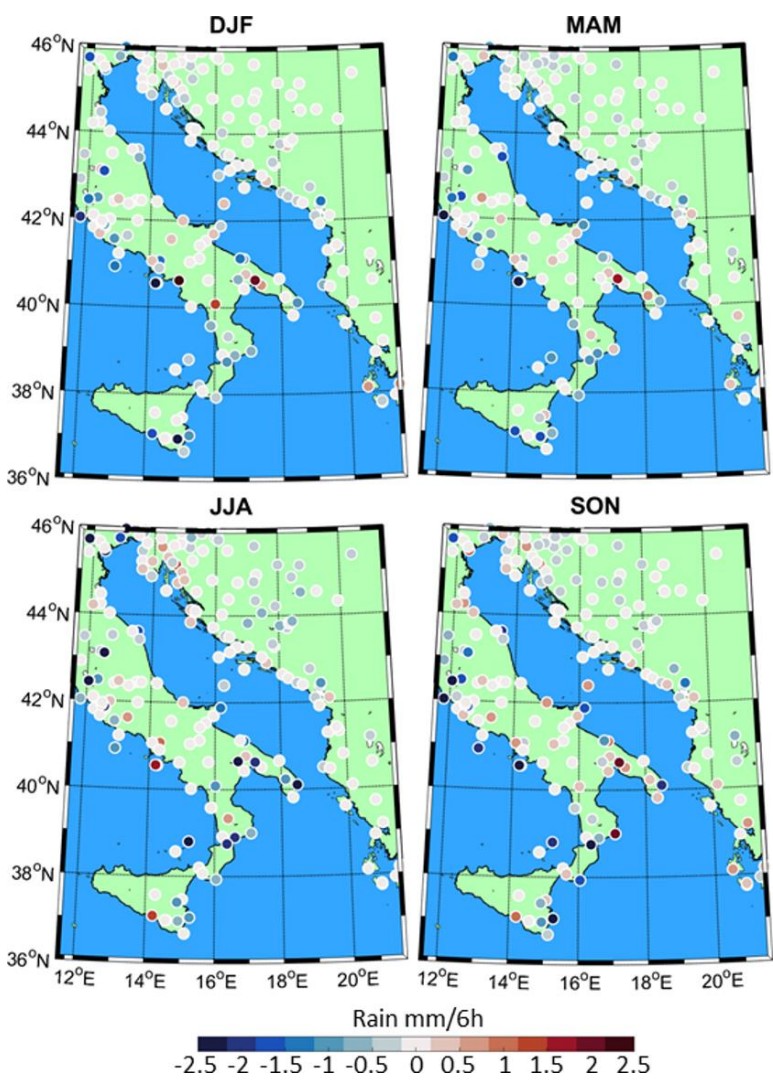

840

**Figure 12. As in Fig. 9 but for the mean 6-hourly rain bias.**



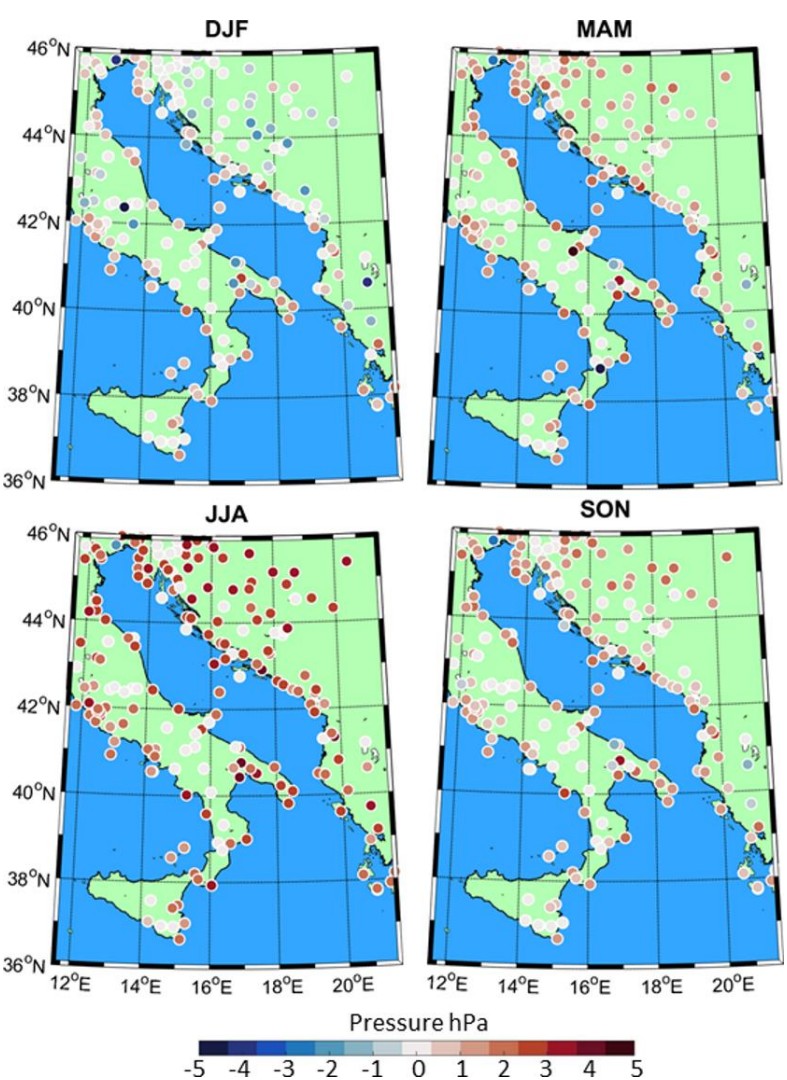

**Figure 13. As in Fig. 9 but for the hourly pressure bias.**



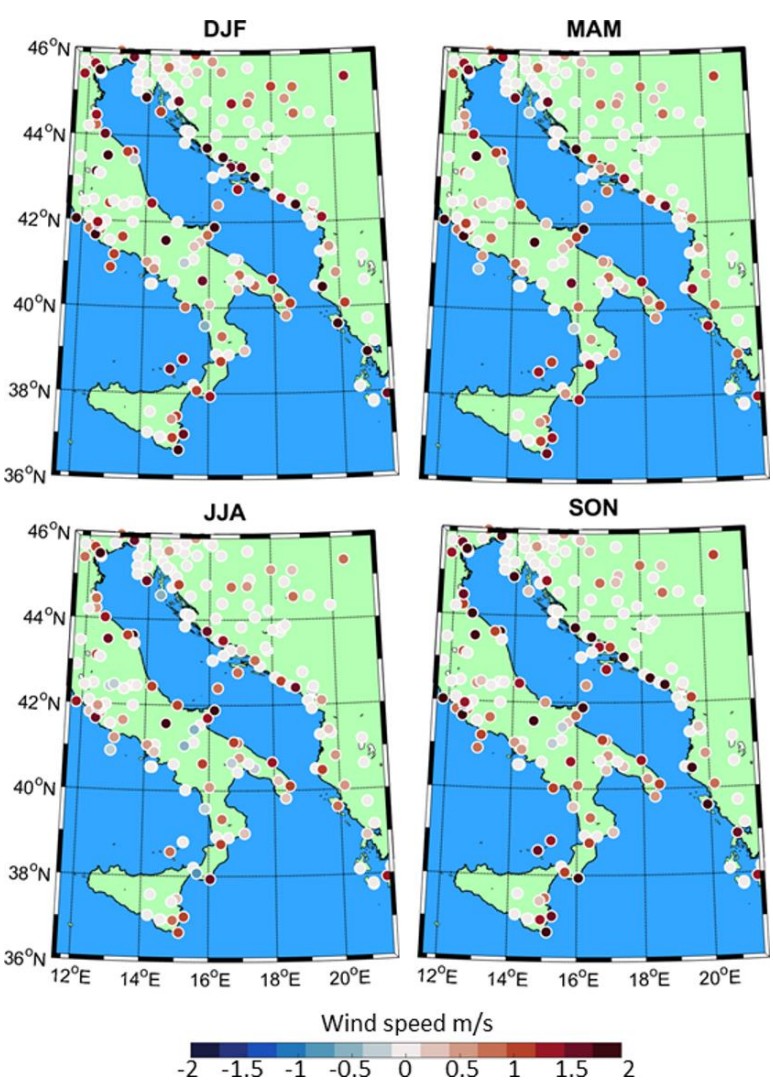

845 **Figure 14. As in Fig. 9 but for the hourly wind speed bias.**

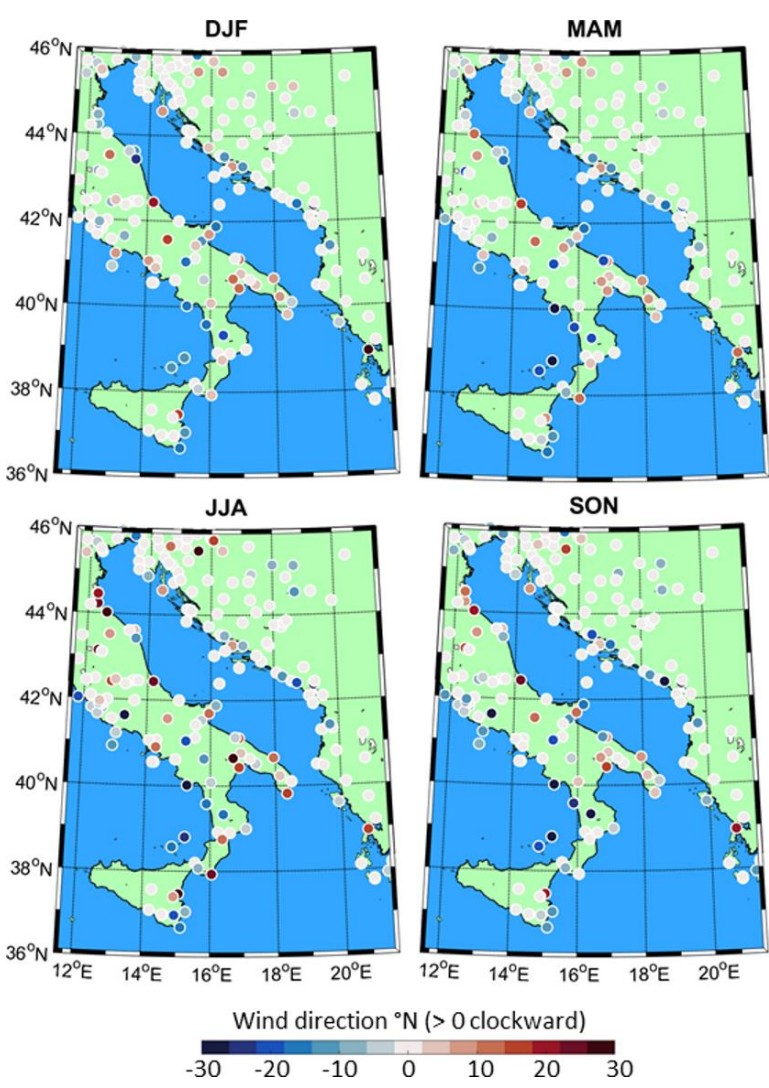

**Figure 15. As in Fig. 9 but for the hourly wind direction bias.**



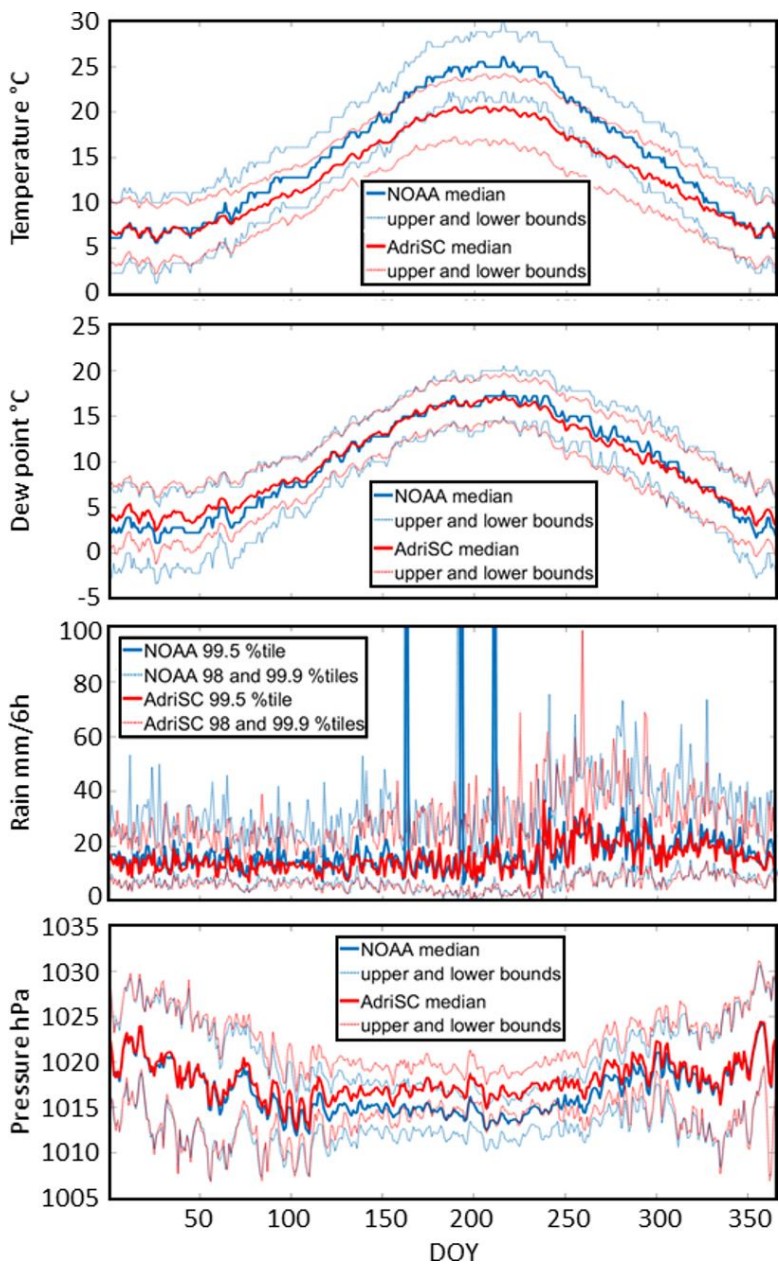

**Figure 16. Daily climatology of the median temperature, median dew point, extreme rain (i.e. 99.5th percentile), median pressure and their variabilities (i.e. upper and lower bounds defined as ±MAD or 98th and 99.9th percentiles for the rain) for both AdriSC WRF 3-km model results and NOAA measurements over the entire domain and 1987-2017 period. The abbreviation DOY stands for Day-Of -Year.**



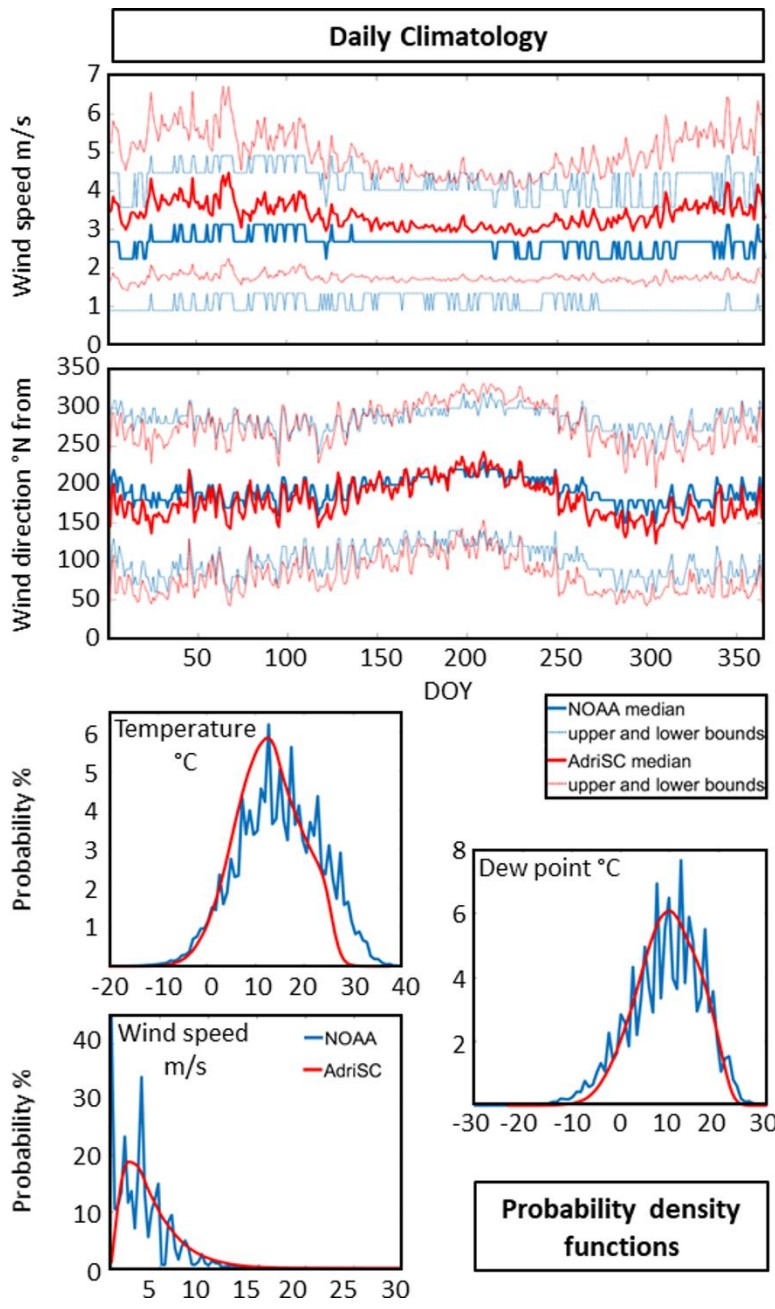

**Figure 17. Temperature, dew point and wind speed probability density functions (top three panels) as well as daily climatology of median wind speed and direction and associated variabilities as upper and lower bounds defined as ±MAD (bottom two panels) derived from the NOAA stations measurements and the corresponding AdriSC WRF 3-km model results over the entire domain and 1987-2017 period. The abbreviation DOY stands for Day-Of -Year.**



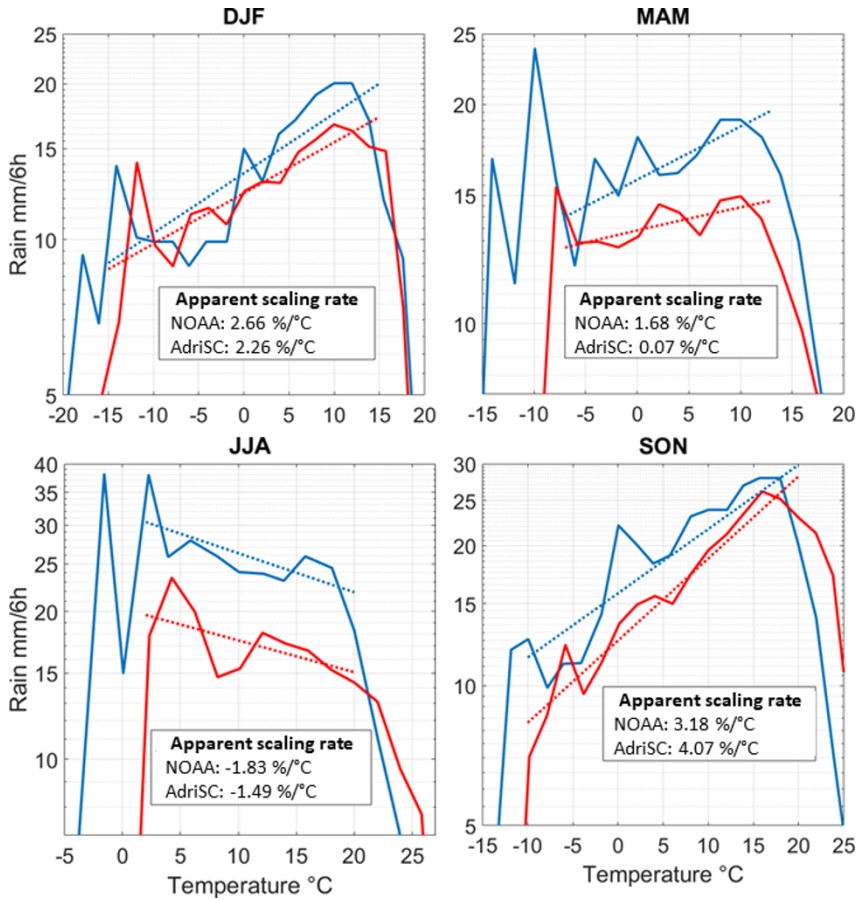

**Figure 18. Seasonal apparent scaling rates derived for both the AdriSC WRF 3-km model (i.e. AdriSC, in red) and the observations (i.e. NOAA, in blue) from the dataset of the 251 NOAA stations and defined as the linear relationship between the logarithm of the extreme precipitations (i.e. 99th percentile) and the 2 m temperatures.**



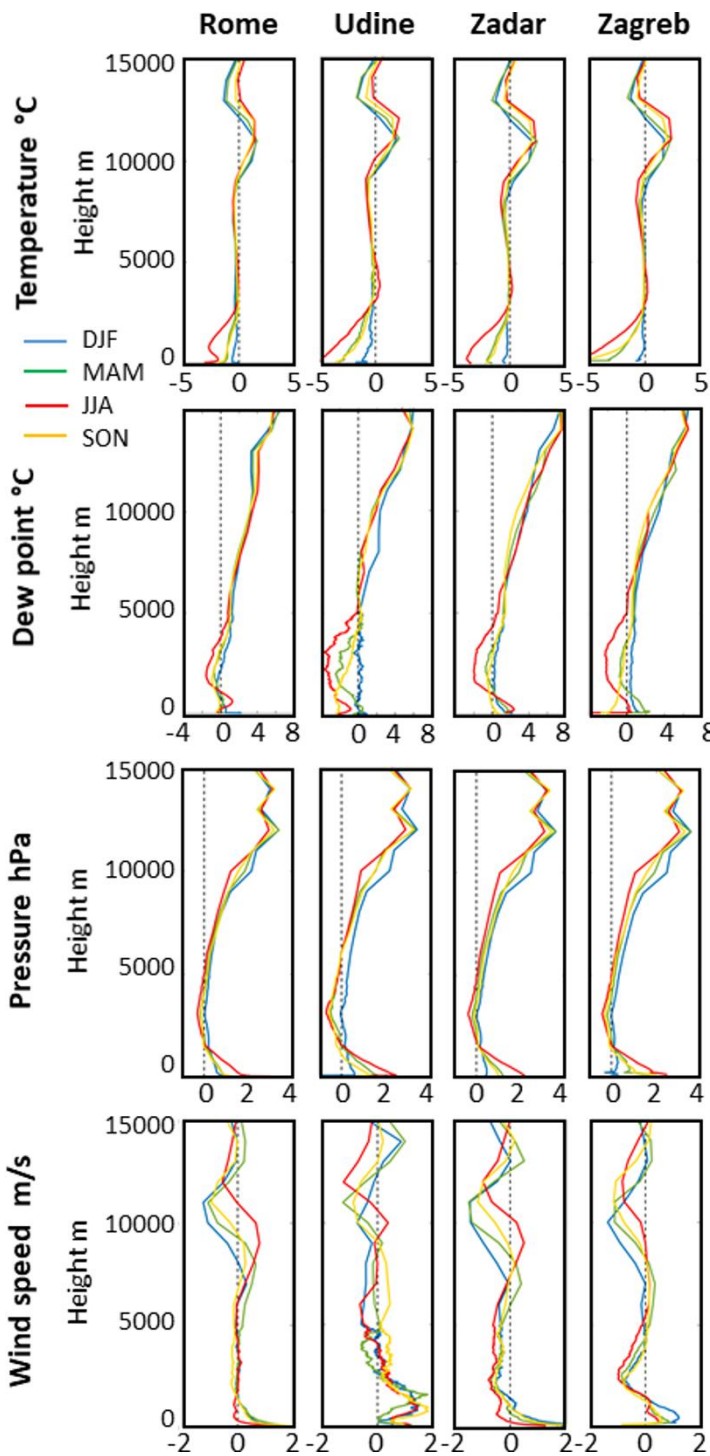

**Figure 19. Seasonal variations of the median of the temperature, dew point, pressure and wind speed biases between the AdriSC WRF 3-km model and the sounding measurements between the surface and 15 km of height for four different locations (Rome, Udine, Zadar and Zagreb).**





| | Atmosphere | | Ocean | |
|---|---|---|---|---|
| Models | WRF | | ROMS | |
| Number of domains | 2 | | 2 | |
| Horizontal resolution | 15 km | 3 km | 3 km | 1 km |
| Vertical resolution | 58 levels | | 35 levels | |
| Time step | 60 s | 12 s | 150 s | 50 s |
| Initial and boundary Conditions | ERA-Interim | | MEDSEA | |
| 31-year period | 1987-2017 | | | |
| Frequency of outputs | Hourly | | | |

**Table 1. Summary of the AdriSC climate component main features for the evaluation run.**

| Variables | NOAA stations Hgt.* (m) | NOAA stations Freq.** (h) | NOAA stations #*** $10^6$ | E-OBS Hgt. (m) | E-OBS Freq. (h) | E-OBS # $10^6$ | CCMP Hgt. (m) | CCMP Freq. (h) | CCMP # $10^6$ | TRMM Hgt. (m) | TRMM Freq. (h) | TRMM # $10^6$ | UWYO Soundings Hgt. (km) | UWYO Soundings Freq. (h) | UWYO Soundings # $10^6$ |
|---|---|---|---|---|---|---|---|---|---|---|---|---|---|---|---|
| Temperature | 2 | 1 | 19 | 2 | 24 | 51 | | | | | | | 0-15 | 12 | 4 |
| Dew point | 2 | 1 | 19 | | | | | | | | | | 0-15 | 12 | 3 |
| Pressure | msl**** | 1 | 8 | msl | 24 | 51 | | | | | | | 0-15 | 12 | 4 |
| Rain | 0 | 6 | 2 | 0 | 24 | 51 | | | | 0 | 24 | 15 | | | |
| Wind speed | 10 | 1 | 13 | | | | 10 | 6 | 90 | | | | 0-15 | 12 | 4 |
| Wind direction | 10 | 1 | 10 | | | | 10 | 6 | 90 | | | | 0-15 | 12 | 4 |

*height    **frequency    ***number of records    ****mean sea-level

870   **Table 2. Height, frequency and number of records of the different variables from the 5 datasets used for the evaluation of the AdriSC WRF 3-km model over the 1987-2017 period.**