# Peer review of "Figure S1. MAD of the E-OBS daily mean temperature, rain and mean sea-level pressure datasets over the land (left panels) as well as MAD (right panels) of the daily temperature, rain and mean sea-level biases between AdriSC WRF 3-km model results and E-OBS datasets over the land during the 1987-2017"

_Geoscientific Model Development, 2021_

## Author Response (AR1)

**Response to Reviewer #1 comments**

**General comments:**

*In the paper entitled "Performance of the Adriatic Sea and Coast (AdriSC) climate component – a COAWST V3.3-based coupled atmosphere-ocean modelling suite: atmospheric part" the Authors introduce and analyze the performances of the atmospheric compartment of the AdriSC coupled model. The paper is interesting and opens the door to some new important development in the Mediterranean modeling community.*

Response: Thank you very much for your detailed and constructive review.

*However, I'm little bit puzzled about some points that I summarize here:*

1. *The Authors do a very robust validation of the atmospheric variables using several datasets. But in the text it looks like that some of them are not very reliable. And so I asked: why do not You use another dataset? I never saw papers using EOBS for the Sea Level pressure. Why do not you use ERA-interim which assimilates Sea level pressure and thus it can be considered more robust for the validation?*

Response: The authors have two main reasons for not using ERA-I for the evaluation of their model: (1) ERA-I is a re-analysis product at 0.75° (thus not the observational data product, but just assimilates observations through 4D-Var assimilation procedures), while E-OBS is a product directly derived from observations at 0.1°, and (2) ERA-I is already used to force the boundaries of the AdriSC atmospheric model, which means that the evaluation would not be done with independent observations/dataset.

Additionally, several studies evaluating climate models have been using E-OBS sea-level pressure, such as Kotlarski et al. (2014) which evaluated the full EURO-CORDEX ensemble.

Finally, as stated in the introduction, this evaluation article was also used to test the reliability of the available observations in the Adriatic region. The fact that E-OBS sea-level pressure product clearly presents some problems was discovered thanks to the analysis performed in this study. This E-OBS problem was reported to the data creator and will be fixed in the next release of the product, as it was found that it is linked to a conversion of air pressure to mean sea-level of two stations along the Adriatic coast. The following sentence was added in the manuscript: *"In fact, the issue – linked to a wrong conversion of the air pressure data to mean sea-level at two different stations along the Adriatic coast – has been reported to the data creator and will be fixed in the next release of the E-OBS products."*

2. *The text is full of numbers and thus for a reader sometimes it is difficult to follow the discussion. I think the Authors should avoid such an inflation of numbers and try to establish some take-home messages focusing on the most important biases.*

**Response**: As evaluation articles generally aim to quantify the biases of the numerical models compare to observations, the authors recognize that they tend to be, by nature, extremely repetitive, full of numbers and somewhat tedious to read. For that reason, the authors placed summary paragraphs at the end of each demanding evaluation subsection (Sect. 3.2 to 3.4), which are summarizing the most important results coming from the respective evaluations. Further, the authors put more accent to take-home messages placed in summary and perspectives section (e.g. see three highlighted points in the first paragraph).

3. *Information about the coupled system are missed: for example, if you consider a spin-procedure or not for the ocean part? Any information about the river (which are important in the thermohaline circulation of the northern part of the basin)?*

**Response**: If not being insisted by the reviewer and the editor, the authors would like to provide the information concerning the set-up of the ocean model in the article by Pranić et al. (2021), soon (in less than a month) to be submitted to Geoscientific Model Development. This article will be dedicated to the presentation and the full evaluation of the ocean component of the AdriSC climate model. In order to avoid auto-plagiarism, the authors are in favour not to duplicate the information that will be presented in this future article. This is clearly noted in the first paragraph of Section 2.1.2.

4. *The most important point: the Authors list very carefully the biases with respect to the observations. But as far as I see the discussions of the source of these biases are missed. I think that a possible user of the modeling tool should be aware of the existence of the biases and the respective sources. This should also help the improvement of the performances in the future. The biases observed are related to convective scheme? Boundary layer? Boundary conditions? Land surface scheme*

**Response**: During the analysis of the results, the authors have tried to link biases to either problems with the observations *per se* or the physics of the WRF 3-km model. Some examples related to the WRF model configuration are presented hereafter:

- *"These results are following the work of Varga and Breuer (2020) who, for a 1-year long period, studied the sensitivity of simulated 2 m temperature to different WRF 10-km configurations over a domain which partially cover the Adriatic basin. Specifically, they found that, for any WRF configuration, the spatial distributions of the annual mean temperature bias relative to the E-OBS dataset present a general underestimation of about -4.0 to -3.0 °C."*
- *"This boundary effect is linked to the fact that the WRF 3-km model which resolves some of the small-scale convective clouds, is nested into the coarser WRF 15-km domain for which the Kain-Fritch cumulus parameterization (Kain, 2004) is used."*
- *"Similar results were found by Varga and Breuer (2020) for a WRF model using the same physics than the AdriSC WRF 3-km but coarser horizontal (10-km) and vertical (31 levels) resolutions, particularly in summer (i.e. biases down to -5.4 °C in July) but also for all the other seasons (i.e. bias of -4.5 °C annually). They also demonstrate that the temperature bias can be largely reduce by using other numerical scheme for the planetary boundary and surface layers than the ones used in this study."*

The authors, however, recognize that the message about the problems with the WRF set-up configuration may have been too diluted in the article and added the following paragraph in the summary and perspectives (Section 4):

*"Overall, the presented work highlighted three important points. First, the AdriSC WRF 3-km model demonstrates some skill to represent the climate variables, with the exception of the summer temperatures systematically underestimated by up to 5 °C over the entire domain. Second, some of the quantified biases are directly linked to the physics set-up of the AdriSC WRF 3-km model. For example, as the AdriSC WRF 3-km model resolves some of the small-scale convective clouds, the boundary effects seen in the spatial rain biases are linked to the Kain-Fritch cumulus parametrization used in the mother grid (i.e. the AdriSC WRF 15-km model). More importantly, the summer temperature biases found over the entire 3-km Adriatic-Ionian domain can definitely be linked to the choice of the MYJ and Eta numerical schemes (Janjić, 1994) used for the planetary boundary and surface layers, respectively. Indeed, Varga and Breuer (2020) have recently demonstrated that replacing the MYJ scheme with the University of Washington (UW; Bretherton and Park, 2009) parameterization could improve the representation of the temperature over their domain partially covering the Adriatic region."*

*To summarize I think that the paper deserves to be published in GMD but only after some major revisions which address all these points.*

**Response**: Thanks again for your review, the authors hope they now have clarified all the concerns raised by the reviewer.

**Specific comments:**

- *Line 24-34: The Authors correctly list a series of RCMs developed in the framework of Cordex-initiative. However, they do not report the Med-Cordex initiative that focus specifically on the Mediterranean region (which also includes the Adriatic basin which is the focus of their work). I think the Authors in this introduction should focus more on the Med-Cordex initiative eventually discussing the development of regional coupled system in this framework (see for example Sevault et al., 2014; Ruti et al., 2016; Somot et al., 2018; Reale et al., 2020; Sein et al., 2020) than on the global Cordex initiative. Moreover, I see the message in the sentence "RCMs…land-sea contrast" but the sentence , I think, is misleading as RCMs are specifically developed to better resolve topography etc.. Additionally, the coupling between ocean and atmosphere works well (sometimes) also in the open ocean areas not only in the coastal areas. To summarize I would reformulate the paragraph*

**Response**: The authors agree that Med-CORDEX initiative should be mentioned. The following sentence was thus added:

*"Specifically, in the Mediterranean Sea, several RCMs have been developed within the Med-CORDEX initiative (e.g. Sevault et al., 2014; Ruti et al., 2016; Somot et al., 2018; Reale et al., 2020; Sein et al., 2020)."*

However, the authors stand by their statement that RCMs are not designed to represent extreme events due to their relatively coarse resolution, which is described in many details

in Prein et al. (2015) but also mentioned in the CODEX FPS 2020 call which link was given in the text.

- *Fig.1 I would show in first panel also the bathymetry of the Adriatic Sea (which you discussed in the text and it is quite interesting) because the coupled model has an ocean compartment*

**Response**: Accepted. Figure 1 was amended with the bathymetry added on the same panel than the topography.

- Line 40-44 As far as I remember the Adriatic Ionian Sea interaction (BioS ) is driven by dense water formation in the Southern Adriatic and not by the bora wind. Deep water formation in the Northern Adriatic becomes important only in case of extreme events (see winter 2012, Gacic et al., 2012). I would rephrase the sentence.

**Response**: There is an extensive literature on the dense water in the southern Adriatic that is also driven by exceptional bora wind forcing, through open-convection processes (while being generated by shelf cooling in the northern Adriatic). The difference between these two dense water formation sites are in both thermohaline characteristics and volumes of the generated water masses. The text was modified accordingly as:
"*Additionally, orographically-driven extreme windstorms mostly from the north-eastern direction (i.e. the so-called bora winds; Brzović and Strelec Mahović, 1999; Grisogono and Belušić 2009) are known to strongly influence the annual dense water budget in the Adriatic Sea. The dense waters are formed on both northern Adriatic shelf (through shallow-water cooling, Janeković et al., 2014) and in deep southern Adriatic (through open-ocean convection, Gačić et al., 2002) and are a driver of interannual to decadal thermohaline and biogeochemical variability between the Adriatic and the northern Ionian seas (Roether and Schlitzer, 1991; Gačić et al., 2010; Bensi et al., 2013; Batistić et al., 2014).*"

- *Line 53 "to" proper not "for" proper.*

**Response**: Accepted. For was replaced with *"to"*.

- *Line 56 Why do you choose 1987-2017? (Because of the ocean MEDSEA reanalysis?)*

**Response**: Yes. The text was amended with the following addition:
*"It should be noted that, in 2018 when the climate model was set-up, the 1987-2017 period was chosen due to the availability of reliable daily ocean re-analysis in the Mediterranean Sea."*

- *Line 57-77 I think this part should go in data and methods. In the Introduction the reader is more interested in learning about the scope of the work and its structure*

**Response**: The authors think that discussion about the availability and reliability of the observations can be part of the introduction as evaluation articles fully rely on the quality of the observations used for comparison with model results. Observations, like models, come with intrinsic shortcomings that should be discussed and acknowledged in the introduction,

as potentially affecting the overall quality of the evaluation article. Line 57-77 thus remains in the introduction. However, as also highlighted by Reviewer #2, the framework of the article may not have been clearly presented and the following paragraph was added:
*"The following study solely assesses the skill of the AdriSC atmospheric kilometre-scale model while the evaluation of the AdriSC ocean coastal model is done separately. It also is, as suggested by Massonnet et al. (2016), a bidirectional exercise evaluating both the kilometre-scale AdriSC atmospheric model and the freely available observations retrieved, in the Adriatic basin, from in situ measurements, gridded datasets and remote-sensing products. The presented work thus aims at answering the following questions: What are the strengths and shortcomings of the AdriSC atmospheric model depending on the evaluated essential climate variables and how are they related to the physical set-up of the model? Are the skills of the newly developed climate model similar at the daily and hourly time-scales? How the performance of the kilometre-scale atmospheric model compare to the RCMs set-up within the CORDEX community? What is the quality and the reliability of the freely available observations in the Adriatic region?"*

- *Section 2.1.3: this section should go before section 2.1.2 that describes the portal*

**Response**: Accepted. The two sections were inverted.

- *Line 140 Which resolution ERA-interim has? 1.5/0.75°? The citation for ERA-interim I think should be Dee et al. 2011*

**Response**: Accepted. The following was added:
*"ERA-Interim reanalysis fields at 0.75° resolution (Dee et al., 2011; Balsamo et al., 2015)"*

- *Line 145-146 You should report that the reanalysis is a CMEMS product*

**Response**: Accepted. The following was added:
*"MEDSEA re-analysis from the Copernicus Marine Environment Monitoring Service (CMEMS; Simoncelli et al., 2014)"*

- *Line 158-159: I would cite some works using E-OBS*

**Response**: Accepted. The following references were added:
*"(e.g. Kotlarski et al., 2014; Varga and Breuer, 2020)"*

- *Line 240-246 I do not see the Authors'point here. You are comparing hourly data with hourly data and then you have a good sampling and the correlation is good. If you compare hourly data with 6-hourly or daily data maybe you are suffering of some oversampling/undersampling due to the frequency considered. I would remove this sentence otherwise the Authors should explain better with some examples their findings.*

**Response**: The point of the authors is that timing of extreme events is not expected to be reproduced with climate models. Consequently, it is expected that evaluations made with daily averages should be better than the ones made with hourly data. For example, if a

modelled storm is shifted by some few hours it can still be simulated in the same day than it was observed. However, few hours' shift will be seen as big differences if the comparison is made hourly. The following explanation was added:
*"(e.g. a modelled historical storm shifted by few hours compared to the reality can still be synchronized with the daily averaged observations but would definitely generate big biases if compared to hourly measurements)"*

- *Fig.4 Why do not you show the first, 25, 75, 99 percentile in this order?*

**Response**: As the percentile analysis is done on the biases which can be positive or negative, the choice of the order is only linked to how extreme are the percentiles: 25th and 75th percentiles represent "medium" extremes (negative and positive, respectively) while 1st and 99th percentiles represent "extreme" extremes (negative and positive, respectively).

- *Section 3.2 As I said in the general comments the Authors should describe the possible sources of the biases observed. In this way a reader or a possible future user of the data can appreciate the possible limits of the tool available. For example 8 C in 99th percentile is quite high. Do you have an explanation for that? Boundary layer scheme or short wave radiation overestimation related to ERA-interim?*

**Response**: The fact that the cold bias is linked to the boundary layer scheme is discussed later in the article, once all the analyses with all the datasets are performed:
*Lines 439-442: "Similar results were found by Varga and Breuer (2020) for a WRF model using the same physics than the AdriSC WRF 3-km but coarser horizontal (10-km) and vertical (31 levels) resolutions, particularly in summer (i.e. biases down to -5.4 °C in July) but also for all the other seasons (i.e. bias of -4.5 °C annually). They also demonstrate that the temperature bias can be largely reduce by using other numerical scheme for the planetary boundary and surface layers than the ones used in this study."*

Concerning the 8 °C bias obtained for the 99th percentile, the extreme events (e.g. bora events which generate extreme cooling) can be delayed for few hours compared to the observed data and thus extreme cold/heat can be shifted to a previous/next day which would definitely create a really high bias. The 99th percentile is the "extreme" extreme, consequently, it can be safely said that 8 °C is nearly the maximum positive bias in the model during the entire 1987-2017 period.

- *Fig.5 I would revert the color bar showing in blue the peak of the precipitation or the overestimation of precipitation and the red the drier conditions or the underestimation. Around the zero the color should be white not gray.*

**Response**: Figures 4 to 9 have been designed with the same colour bars as they present the same kind of data (i.e. median observations and median, 1st, 25th, 75th and 99th percentiles of the bias). The authors truly believe it would be far more confusing for the readers if the colour bar for the rain would be suddenly reverted and changed. The authors thus kept the same colour bar for Figure 5 than for the other figures (4, 6 to 9). Additionally, the eggshell colour for the values around zero has been chosen instead of white in order to remind the readers that bias is rarely exactly zero but more often close to zero.

- *Line 325-326 as in the general comment if you think that the dataset CCMP is not very reliable as reference dataset (also EOBS in the case of Mean SLP) why do not use another dataset just for the comparison (ERA-interim reanalysis could be a good reference as for MSLP)?*

**Response**: As explained in the response to general comments, ERA-I is definitely not a great dataset for comparison with WRF 3-km in the Adriatic region. CCMP resolution is 0.25° (similarly to ERA5 re-analysis) and mostly derived from observations (in situ and remote sensing). Clearly, the resolution is not good enough to capture extreme events such as bora storms. However, the authors wanted to check whether or not the integration of observations would compensate the lack of resolution as no such results have been published before in the Adriatic region. In the opinion of the authors, the fact that CCMP may not be accurate enough in the Adriatic region is per se a result of this article.

- *Line 375 1 C bias is not slightly. please remove "slightly"*

**Response**: Accepted. *"Slightly"* was removed.

- *Fig.12 -Fig.15 I would color both land and sea in white and leaves in black only the coastlines: the marker line of each dot can be also done in black*

**Response**: The suggestion of the reviewer was implemented in Figure R1 (see below) for the 2 m air temperature, as an example. The authors believe that the obtained figure looks extremely empty and is overall more difficult to read due to a lack of contrast which is not present in the original figures. Consequently, the authors kept Fig. 12 to 15 as presented in the original version of the article.

- *Line 558-560 (and after) I would be a little bit careful with that. As far as I remember Theocharis et al., 2014, Reale et al., 2016 and Reale et al., 2017 discussed the necessity of the inclusion of the Aegean Sea in the Adriatic Ionian to explain the BIOS variability…I should discuss that in the sentence. Moreover, does it mean that you plan to include also the Aegean Sea in your modeling domain? Do you expect that the MEDSEA (which provide the BCs in the ocean model) reanalysis include the BiOS signal? Did you check that?*

**Response**: Yes, the authors have checked that the BiOS is reproduced by the MEDSEA re-analysis. Gačić et al. (2011) have demonstrated that the BiOS is well described with the decadal change of sign of one of the main components of the EOF derived from sea-surface height. Figure R2 (see below) is indeed showing the EOF analysis (3 first components) of the sea-surface height extracted for 1987-2017 period within the Ionian Sea. The BiOS is also seen by the ROMS 3-km ocean model used in the AdriSC climate model and the results will be soon submitted by Pranić et al. (2021) to GMD. However, the authors would agree with the reviewer that the BiOS is located a bit too close to the southern open boundary of the AdriSC WRF/ROMS 3-km grids and will be influenced by the MEDSEA forcing in the AdriSC climate model. Nevertheless, the authors also believe that they can still use the 31-year AdriSC 3-km results to look at the drivers of the BiOS particularly to confirm whether this driver is the dense water coming from the Adriatic Sea.

The sentence was reformulated as follow:
*"Additionally, as the MEDSEA re-analysis captures the BiOS signal within the Ionian Sea (e.g. Pinardi et al., 2015), the inclusion of the Aegean Sea suggested by Reale et al. (2017) is not necessary if MEDSEA is used as a forcing. Consequently, the AdriSC climate model has also been developed with the aim to expand the knowledge on whether the Adriatic dense waters travelling towards the Ionian Sea can be an important driver of the BiOS."*

- *Line 570-573 I found interesting the idea to apply your model to simulate future scenarios. But as I've seen from the text to run your model for 31 years (without spin-up) you need at least 18 months. For a longer run how would you deal with a simulated period of 100years? What about the spin-up of the ocean part?*

**Response**: Actually, the idea is to use the Pseudo-Global Warming (PGW) methodology originally developed by the atmospheric community to run kilometer-scale climate models and recently extended to coupled atmosphere-ocean models in Denamiel et al. (2020a) as explained in Section 2.1.1. In more details, the PGW method is based on adding to the 31-year forcing used in evaluation mode (here, ERA-I and MEDSEA) a climatological change derived from other RCM results. For the AdriSC climate model, the LMDZ4-NEMOMED8 RCM model (Hourdin et al. 2006; Beuvier et al. 2010) forced by the IPSL-CM5A-MR GCM model (simulations r1i1p1) have been used to extract the RCP 8.5 climatological changes for the already running AdriSC simulation (i.e. more than 20 years of results under RCP 8.5 scenario via the PGW method are already available). As for the evaluation run, the warm-up period of the AdriSC RCP 8.5 climate run is reduced to only two-months (November and December 1986 forced with monthly MEDSEA re-analysis). On one hand, it can be argued that this 2-months' warm-up may not be enough for the ocean to adjust to the PGW RCP 8.5 forcing but, on the other hand, the Adriatic domain is far smaller than the entire Mediterranean Sea and the PGW method only add climatological changes to the MEDSEA and ERA-I re-analyses, which should also speed up the convergence to steady-state of the AdriSC model.

Finally, as the AdriSC coupled atmosphere-ocean kilometer-scale climate model is the first of its kind (at least to the knowledge of the authors) to be run for the future climate using the PGW method, the authors hope the reviewer will appreciate that not all the answers about the reliability of such a future scenario run can yet be known and that only the full analysis of the finished RCP 8.5 run (expected in autumn 2021) will provide undeniable proof of the potential failure/success of the approach.

[Figure]

Figure R1. Seasonal variations of the median hourly temperature bias between AdriSC WRF 3-km model results and NOAA land station measurements during winter (DJF), spring (MAM), summer (JJA) and autumn (SON) for the 1987-2017 period.

[Figure]

Figure R2. EOF analysis (three first components) of the sea-surface height extracted from MEDSEA for the 1987-2017 period.

**Response to Reviewer #2 comments**

**General/mayor comment:**

*This paper presents an interesting study about the validation of the coupled atmosphere-ocean Adriatic Sea and Coast (AdriSC) climate model over the Adriatic. The topic is relevant, the paper is well organized and written relatively clearly. However, I suggest some corrections in the manuscript and minor revisions.*

Response: Thank you very much for your encouraging comments and helpful review.

**Detailed comments in text:**

*1) Figure 1 Please replace Dinaric Alps with Dinarides*

Response: Accepted. Dinaric Alps was changed with Dinarides in Figure 1.

*2) Page 2, line 39; please replace … located in the northeast...into ... along the Adtiatic coast... or delete*

Response: Accepted. Located in the northeast was replaced with "along the northeast coastline"

*3) Page 2, line41; please replace … in the northern Adriatic ... into ...mostly from the north-eastern direction*

Response: Accepted. In the northern Adriatic was replaced *with* "from the north-eastern direction"

*4) Page 3, line 75: It would need to be better formulated in the introduction the main research question and aims.*

Response: The paragraph was re-written and extended in the text as follow:
"The following study solely assesses the skill of the AdriSC atmospheric kilometre-scale model while the evaluation of the AdriSC ocean coastal model is done separately. It also is, as suggested by Massonnet et al. (2016), a bidirectional exercise evaluating both the kilometre-scale AdriSC atmospheric model and the freely available observations retrieved, in the Adriatic basin, from in situ measurements, gridded datasets and remote-sensing products. The presented work thus aims at answering the following questions: What are the strengths and shortcomings of the AdriSC atmospheric model depending on the evaluated essential climate variables and how are they related to the physical set-up of the model? Are the skills of the newly developed climate model similar at the daily and hourly time-scales? How the performance of the kilometre-scale atmospheric model compare to the

RCMs set-up within the CORDEX community? What is the quality and the reliability of the freely available observations in the Adriatic region?"

*5) Page 7, line 199: Please correct or explain a part of the sentence due to repetition...such as median (or mean for the rain) and Median (Mean for rain).*

**Response**: The paragraph was reformulated as follow:
"The biases are analysed in space with statistical quantities such as median and Median Absolute Deviation (MAD) as well as 1st, 25th, 75th and 99th percentiles. In this study, in order to obtain more robust statistics for the chosen geophysical quantities which are likely to be heavy tailed due to extreme conditions, the use of median and MAD is preferred to the mean and standard deviation preconized for normal distributions. However, despite having a heavy tailed distribution, the rain is not a continuous quantity – i.e. occurrences of rain in the Adriatic region are low and the median is likely to be close to zero. Consequently, the mean and Mean Absolute Deviation (also MAD) are used for the statistical analysis of the rain instead of the median and Median Absolute Deviation."

*6) Page 8, lines 222, 224, 229 & Page 9, line 258... (and further in the text): Please replace the Dinaric Alps with the Dinarides*

**Response**: Dinaric Alps was changed with Dinarides for the 10 occurrences used in the text.

*7) Page 8, line 240; How was done a comparison between radiosondes with the model? On lines 193-194 it can be understood that the model data from the sigma level were interpolated to each radiosonde with a different number of levels and comparisons were made on all isobaric surfaces/levels (standard and significant). Do you have some explanation about the the best statistics for the UWYO soundings?*
*Presumably, since the comparison is made by height and the influence of the lowest layer is less represented in relation to the middle and higher troposphere where synoptic forcing dominates (and climate models better), the matching is good.*

**Response**: Yes, the vertical model results on sigma layers were indeed interpolated on the exact heights of the radiosonde measurements which are having different heights for each time of the measurements, as explained in lines 193-194. For the Taylor plot the comparisons were made for all heights without distinction. It can be argued that an interpolation of the measurements and results could have been made on the regular heights (as done later for the bias comparison). However, the Taylor plot is "only" used to provide a rough estimate of the model performances and, as pointed by the reviewer, the main reason for doing the comparison between model and radiosonde measurements is the fact that climate models do better on the higher troposphere where synoptic forcing dominates than in the surface layers. This is discussed during the analysis of Figure 19.  The following sentence was added in order to clarify the interpretation of the Taylor plot:
"However, it should be noted that most of the sounding measurements are taken in the higher troposphere (i.e. about 90% are above 1 km of height) where synoptic forcing dominates and, hence, where climate models generally perform better than in the surface layer."

*8) Page 10, line 300; The comment is related to the maximium over the mountainous part near the northern edge of the domain. How is this deviation up to 8.5 hPa (just an inaccuracy of the E-OBS base?) interpreted with the assumption that the positive bias is relatively uniform occurred over teh continent/lan area of the northeastern part of the domain.*

**Response**: The positive bias of about 8.5 hPa in the north-eastern part of the domain is in fact mostly covering the Pannonian plain. The authors do not believe it is a problem with the E-OBS product as the same bias is seen on the QC NOAA stations for the median. As seen on the daily climatology extracted from the NOAA stations, the underestimation of the summer 2 m temperatures with the AdriSC model is accompanied by an overestimation of the mean sea-level pressure. As the worst underestimation of the temperatures occur in the Pannonian plain, it makes sense that it is accompanied with the worst overestimation of the mean sea-level pressure while comparing E-OBS and AdriSC climate model. The following sentence was added:
"It should be noted that the largest mean sea-level pressure positive biases (about 7 hPa for the 75th percentile) occur in the Pannonian plain (i.e. north-eastern edge of the domain), where the largest 2 m temperature negative biases (about -8 °C for the 25th percentile) are also located."

*9) Page 11, lines 309-325; Apart from the distribution of numbers itself, how it is possible to interpret the distribution of median values of wind direction along the Adriatic (Fig. 8) in terms of the flow regime? The bora flow could be typical near the coast which changes to the sirocco over the middle of the Adriatic, or not? Wind speed can be treated as a temperature, but wind direction has a problem with a circular wind rose, so e.g. 240-280 ± 40-80 ° can very easily mean both bora (0-90° & 330-360°) and sirocco (90-180°). I suggest that you consider the vector mean as a possible representation of the mean flow field.*
*Be also careful with the way of writing the wind direction 240-280±40-80 °North; It is unclear whether it is referring - to the azimuth or the directions according to the wind rose (also in the Fig. S2). In the later case, this is not correct.*

**Response**: The authors agree that their original analysis of the wind direction was not appropriate. They decided to switch the analysis from quantitative to qualitative and now compare the median, 25th and 75th percentiles for both CCMP and WRF 3-km model separately and not as a bias. Vectors were also added to the plots in order to clarify the direction of the wind (see new Figure 8). Finally, the MAD analysis of the wind direction was removed for both CCMP and NOAA stations analyses. The following description of the new analysis was added to the text:
"Concerning the wind direction, a qualitative comparison shows that median as well as 25th and 75th percentiles are similar for the CCMP products and the AdriSC WRF 3-km model within the Adriatic Sea, while the biggest differences are seen within the Ionian and Tyrrhenian seas where the AdriSC WRF 3-km model systematically shifts the directions by 40-120° anticlockwise. In more detail, in the Adriatic Sea, the wind is blowing from: (a) 220-280 °North along the Italian coast and 80-120 °North along the eastern coast for the median, (b) 40-60 °North in the northern Adriatic as well as along the eastern coast and 140-180 °North along the southern Italian coast for the 25th percentile, and, finally, (c) 200-240 °N in the northern Adriatic as well as along the eastern coast and 300-360 °North in the rest of the Adriatic Sea for the 75th percentile. However, it should be noted that the wind

directions are much more homogeneous for the CCMP product than for the AdriSC WRF 3-km results mostly due to both the low spatial resolution and the lack of accuracy along the coasts of the remote sensing data. As an example, the bora – a northern to north-eastern downslope wind associated with speeds of 20.0-30.0 m/s (Grisogono and Belušić, 2009) – is regularly blowing along the northern Adriatic and Croatian littoral areas mostly during winter and spring. The different known bora jets (e.g. Trieste in the northern Adriatic and Senj at about 44.5 °N of latitude) represented by directions lower than 60 °North in the 25th percentile can be clearly seen with the WRF 3-km model but not with the CCMP products, which uniformly see directions typical of bora storms along the entire northern Adriatic and eastern coast. Therefore, the differences in directions associated with an overestimation of the wind speeds in the northern Adriatic, may be linked to the CCMP product and not the inaccuracy of the AdriSC WRF 3-km model."

*10) Page 15, lines 434-435; This argument is completely correct due to the MYJ PBL scheme.*

**Response**: The authors agree.